# A toolbox of IgG subclass-switched recombinant monoclonal antibodies for enhanced multiplex immunolabeling of brain

Nicolas P Andrews[1], Justin X Boeckman[1], Colleen F Manning[1], Joe T Nguyen[2], Hannah Bechtold[2], Camelia Dumitras[1], Belvin Gong[1], Kimberly Nguyen[1], Deborah van der List[1], Karl D Murray[1], JoAnne Engebrecht[2], James S Trimmer[1,3]*

[1]Department of Neurobiology, Physiology and Behavior, University of California, Davis, United States; [2]Department of Molecular and Cellular Biology, University of California, Davis, United States; [3]Department of Physiology and Membrane Biology, University of California, Davis, United States

**Abstract** Generating recombinant monoclonal antibodies (R-mAbs) from mAb-producing hybridomas offers numerous advantages that increase the effectiveness, reproducibility, and transparent reporting of research. We report here the generation of a novel resource in the form of a library of recombinant R-mAbs validated for neuroscience research. We cloned immunoglobulin G (IgG) variable domains from cryopreserved hybridoma cells and input them into an integrated pipeline for expression and validation of functional R-mAbs. To improve efficiency over standard protocols, we eliminated aberrant Sp2/0-Ag14 hybridoma-derived variable light transcripts using restriction enzyme treatment. Further, we engineered a plasmid backbone that allows for switching of the IgG subclasses without altering target binding specificity to generate R-mAbs useful in simultaneous multiplex labeling experiments not previously possible. The method was also employed to rescue IgG variable sequences and generate functional R-mAbs from a non-viable cryopreserved hybridoma. All R-mAb sequences and plasmids will be archived and disseminated from open source suppliers.

DOI: https://doi.org/10.7554/eLife.43322.001

*For correspondence: jtrimmer@ucdavis.edu

Competing interests: The authors declare that no competing interests exist.

## Introduction

Antibodies (Abs) are the workhorses of biomedical research. Enhancing the research community's access to extensively validated Abs remains an important goal of antibody developers in both academia and industry (*Taussig et al., 2018*). This has been the topic of a number of recent conferences and commentaries, led to changes in journal practices as related to transparent reporting of antibody-based research including the use of Research Resource Identifiers or RRIDs, and resulted in large-scale NIH and EU Ab development initiatives (*e.g.*, the NIH Common Fund Protein Capture Reagent Program, the EU Affinomics Program). It is widely recognized that production of Abs in a renewable form, such as monoclonal antibodies (mAbs), represents a substantial advance over polyclonal Abs from antisera that are available in finite quantity, and that comprise a heterogeneous and less definable population of Abs (*Harlow and Lane, 1988*; *Greenfield, 2014*; *Busby et al., 2016*). Having mAbs available in recombinant form as recombinant mAbs (R-mAbs) offers numerous additional advantages (*Bradbury and Plückthun, 2015*). Recombinant expression systems allow for the reliable production of an R-mAb not prone to loss by genetic instability of tetra- or hexa-ploid hybridoma cells or other factors (*Andreeff et al., 1985*; *Barnes et al., 2003*). Moreover, unlike

**eLife digest** The immune system fights off disease-causing microbes using antibodies: Y-shaped proteins that each bind to a specific foreign molecule. Indeed, these proteins bind so tightly and so specifically that they can pick out a single target in a complex mixture of different molecules. This property also makes them useful in research. For example, neurobiologists can use antibodies to mark target proteins in thin sections of brain tissue. This reveals their position inside brain cells, helping to link the structure of the brain to the roles the different parts of this structure perform.

To use antibodies in this way, scientists need to be able to produce them in large quantities without losing their target specificity. The most common way to do this is with cells called hybridomas. A hybridoma is a hybrid of an antibody-producing immune cell and a cancer cell, and it has properties of both. From the immune cell, it inherits the genes to make a specific type of antibody. From the cancer cell, it inherits the ability to go on dividing forever. In theory, hybridomas should be immortal antibody factories, but they have some limitations. They are expensive to keep alive, hard to transport between labs, and their genes can be unstable. Problems can creep into their genetic code, halting their growth or changing the targets their antibodies recognize. When this happens, scientists can lose vital research tools.

Instead of keeping the immune cells alive, an alternative approach is to make recombinant antibodies. Rather than store the whole cell, this approach just stores the parts of the genes that encode antibody target-specificity. Andrews et al. set out to convert a valuable toolbox of neuroscience antibodies into recombinant form. This involved copying the antibody genes from a large library of preserved hybridoma cells. However, many hybridomas also carry genes that produce non-functional antibodies. A step in the process removed these DNA sequences, ensuring that only working antibodies made it into the final library. Using frozen cells made it possible to recover antibody genes from hybridoma cells that could no longer grow.

The recombinant DNA sequences provide a permanent record of useful antibodies. Not only does this prevent the loss of research tools, it is also much more shareable than living cells. Modifications to the DNA sequences in the library allow for the use of many antibodies at once. This could help when studying the interactions between different molecules in the brain. Toolkits like these could also make it easier to collaborate, and to reproduce data gathered by different researchers around the world.

DOI: https://doi.org/10.7554/eLife.43322.002

hybridoma cell lines that can express multiple functional heavy and light immunoglobulin chains (*Zack et al., 1995*; *Blatt et al., 1998*; *Bradbury et al., 2018*), recombinant expression ensures production of a single, molecularly defined R-mAb. Recombinant expression can also yield production levels hundreds or even thousands of times higher than is possible with endogenous expressions of mAbs from hybridoma cells (*e.g.*, *Backliwal et al., 2008*; *Fischer et al., 2015*; *Kunert and Reinhart, 2016*). Furthermore, the cloning of R-mAbs provides for permanent, dependable and inexpensive archiving of the R-mAb as plasmid DNA and nucleic acid sequences versus an archiving system relying on expensive cryopreservation of hybridoma cell lines in liquid nitrogen, and their subsequent recovery as viable cell cultures. The conversion of existing mAbs to R-mAbs also allows for more effective dissemination of R-mAbs as plasmids, bacterial stocks or as DNA sequences. However, in spite of these advantages, it remains that many mAbs used for research are produced by hybridomas in culture.

Modern techniques employing Abs as immunolabels [*e.g.*, immunoblotting (IB), immunocytochemistry (ICC) and immunohistochemistry (IHC)] utilize multiplexing of numerous Abs to simultaneously detect multiple targets within a single cell or tissue sample. This allows for direct comparison of the relative amounts and respective characteristics of multiple target molecules within the same sample, while reducing the number of samples needed to accomplish a comprehensive analysis. Typically, the individual primary Abs bound to the sample are detected with secondary antibodies conjugated to distinct reporters, most commonly organic fluorescent dyes, although enzymes, gold particles, etc. are also routinely employed as detection modalities. Multiplex labeling is often accomplished using Abs raised in distinct species, with their subsequent individual detection

accomplished using species-specific secondary antibodies. However, mouse mAbs offer an important advantage for multiplex labeling procedures. Each mouse mAb is a single immunoglobulin (Ig) isotype, generally of the IgG class and if so specifically of a single IgG subclass, most commonly IgG1, IgG2a or IgG2b. Mouse mAbs of distinct IgG subclasses can be robustly, reliably and specifically detected with commercial subclass-specific secondary antibodies, and, as such, can be multiplexed in a manner analogous to Abs from different species (*e.g., Bekele-Arcuri et al., 1996*; *Rhodes et al., 1997*; *Lim et al., 2000*; *Rasband et al., 2001*; *Manning et al., 2012*). One limitation to greater adoption of this approach is that mouse mAb collections generally have an extremely high representation ($\approx$70%) of IgG1 mAbs (*Manning et al., 2012*), which limits the flexibility of multiplex labeling. The conversion of mAbs into R-mAbs allows for their subsequent engineering into forms with properties distinct from their parent mAb, as is routinely done to impact diverse aspects, including target binding affinity, of therapeutic R-mAbs (*Kennedy et al., 2018*). Such engineering could also include switching the heavy chain constant region to impact subclass-specific secondary Ab binding specificity, an approach similar to that used to successfully modify subclass-specific in vivo effector functions (*Wang et al., 2018*).

There are numerous routes to obtaining validated R-mAbs, including their de novo generation from high complexity immune repertoire libraries produced from naïve or immunized animals, combined with selection of target-specific R-mAbs by in vitro display (*Bradbury et al., 2011*). Alternatively, R-mAbs can be generated from existing hybridoma cell lines, which express well-characterized mAbs (*e.g., Crosnier et al., 2010*). Here we undertook conversion of a widely used collection of hybridoma-generated mAbs extensively validated for neuroscience research applications (*Bekele-Arcuri et al., 1996*; *Rhodes and Trimmer, 2006*; *Gong et al., 2016*) into recombinant form. We developed a coherent pipeline of protocols for effective cloning of intact R-mAbs from cryopreserved hybridoma cells and their subsequent validation compared to their parent mAbs. Further, we developed a process to engineer these R-mAbs to IgG subclass-switched forms that provide additional utility for multiplex labeling employing mouse IgG subclass-specific secondary Abs. This approach is feasible and relatively inexpensive for any laboratory that uses standard molecular biology and mammalian cell culture techniques. This approach also represents a reliable method to convert valuable cryopreserved hybridoma collections to the immortalized form of a DNA sequence archive, including hybridomas that are no longer viable in cell culture.

## Results

### Effective cloning of immunoglobulin $V_H$ and $V_L$ regions from cryopreserved hybridomas and generation of R-mAbs

We previously generated a large library of mouse mAbs that have been extensively validated for efficacy and specificity for immunolabeling endogenous target proteins in mammalian brain samples in immunoblotting (IB) and immunohistochemistry (IHC) applications (*Bekele-Arcuri et al., 1996*; *Rhodes and Trimmer, 2006*; *Gong et al., 2016*). Here, we undertook the systematic conversion of a sizable subset of this existing mAb collection to R-mAbs. We developed an innovative pipeline for R-mAb cloning, expression and validation. For the cloning steps, we built upon a previously described method (*Crosnier et al., 2010*; *Müller-Sienerth et al., 2014*) to clone IgG $V_H$ and $V_L$ region sequences, but with the modification that we cloned directly from cryopreserved hybridomas, without the need for their labor-intensive recovery into cell culture. Our overall cloning strategy (*Figure 1A*) employs PCR-mediated amplification to generate IgG $V_H$ and $V_L$ region sequences from hybridoma-derived cDNA, followed by PCR-based fusion of these $V_H$ and $V_L$ regions with a joining fragment that contains a subset of the elements needed for high level expression of R-mAbs from transfected mammalian cell lines. The product of this fusion PCR reaction is then inserted into a plasmid containing the remainder of the elements for propagation in bacteria and expression of intact heavy and light chains in, and secretion of R-mAbs from transfected mammalian cells (*Figure 1B*).

As described in detail in the Materials and methods section, we extracted RNA from cryopreserved hybridoma cell vials, followed by first strand cDNA synthesis and RT-PCR of the IgG $V_H$ and $V_L$ region sequences (*Figure 1A*). We amplified the IgG $V_L$ kappa ($\kappa$) and $V_H$ domain sequences from the hybridoma cDNA templates using a degenerate mouse Ig variable region primer set developed by Gavin Wright and colleagues (*Crosnier et al., 2010*). This expanded on a previously used set

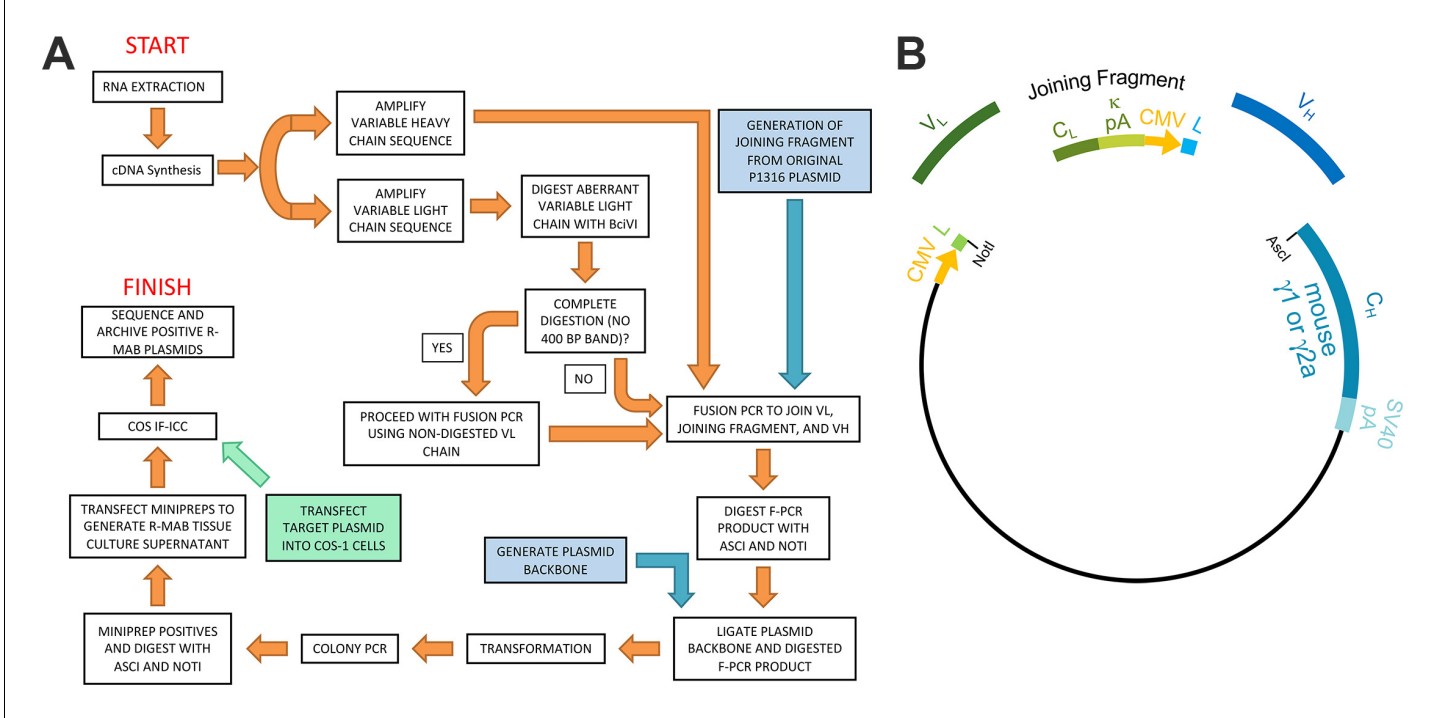

**Figure 1.** Schematic representation of the R-mAb pipeline. (**A**) Schematic of cloning, expression and validation pipeline. Orange steps involve $V_H$ and $V_L$ regions of individual hybridomas, blue steps involve steps involving backbone components, and green step involves expression of target for R-mAb validation. (**B**) Schematic shows the separate elements of the R-mAb expression plasmid involved in coexpression of light (green) and heavy (blue) chains as driven by two CMV promoters (orange). Hybridoma-derived $V_L$ and $V_H$ domain PCR products are fused to a joining fragment comprising a $\kappa$ light chain constant domain ($C_L$) and the $\kappa$ light chain polyA tail sequences ($\kappa$ pA), a CMV promoter for heavy chain expression, and an ER signal/leader sequence (L) for translocation of the heavy chain across the ER membrane. PCR-mediated fusion of these three elements is followed by their insertion into the p1316 plasmid that contains an upstream CMV promoter for light chain expression, and an ER signal/leader sequence (L) for translocation of the light chain across the ER membrane. Downstream of the insert is a heavy chain constant domain ($C_H$) that is either γ1 or γ2a depending on the plasmid, followed by the SV40 polyA tail (SV40 pA).

DOI: https://doi.org/10.7554/eLife.43322.003

(**Krebber et al., 1997**) that at the time of their design recognized 97% and 98% of known functional heavy and kappa (κ) light chain sequences, respectively. Lambda light chains were not targeted for amplification because they constitute only a small percentage of light chains used in mouse immunoglobulins (**Haughton et al., 1978**; **Woloschak and Krco, 1987**). This PCR amplification reliably gave products of the expected 360 base pairs (bp) for both the $V_L$ and $V_H$ domain sequences, examples of which are shown in **Figure 2A** for representative mAbs N59/36 ('N59', anti-NR2B/GRIN2B glutamate receptor) and K39/25 ('K39', anti-Kv2.1/KCNB1 potassium channel).

To permit cloning of both the $V_H$ and $V_L$ regions into a single expression vector, fusion PCR (F-PCR) was performed using as templates the $V_H$ and $V_L$ PCR products, as well as a joining fragment amplified from the P1316 expression plasmid (**Crosnier et al., 2010**; **Müller-Sienerth et al., 2014**) to produce a 2.4 kbp amplicon (**Figure 2B**). The joining fragment (**Figure 1B**) contains kappa light chain constant region sequences and an associated polyadenylation signal, followed by a CMV promoter to drive $V_H$ expression, and a $V_H$ leader sequence (**Crosnier et al., 2010**; **Müller-Sienerth et al., 2014**). The F-PCR reaction products were then treated with NotI and AscI restriction enzymes and purified in preparation for cloning into a NotI/AscI restriction enzyme-digested fragment of the P1316 expression plasmid (**Figure 1B**). Upstream of the NotI cloning site, the digested P1316 fragment contains a CMV promoter to drive light chain expression and a $V_L$ leader sequence, and downstream of the AscI cloning site a mouse IgG1 $C_H$ sequence, and a polyadenylation signal (**Figure 1B**). Clones expressing the full-length IgG expression cassette were identified by colony PCR (**Figure 2C**). Following NotI/AscI restriction digestion to verify the correct insert size (**Figure 2D**), these plasmid clones were subjected to further analysis (**Figure 1A**), including

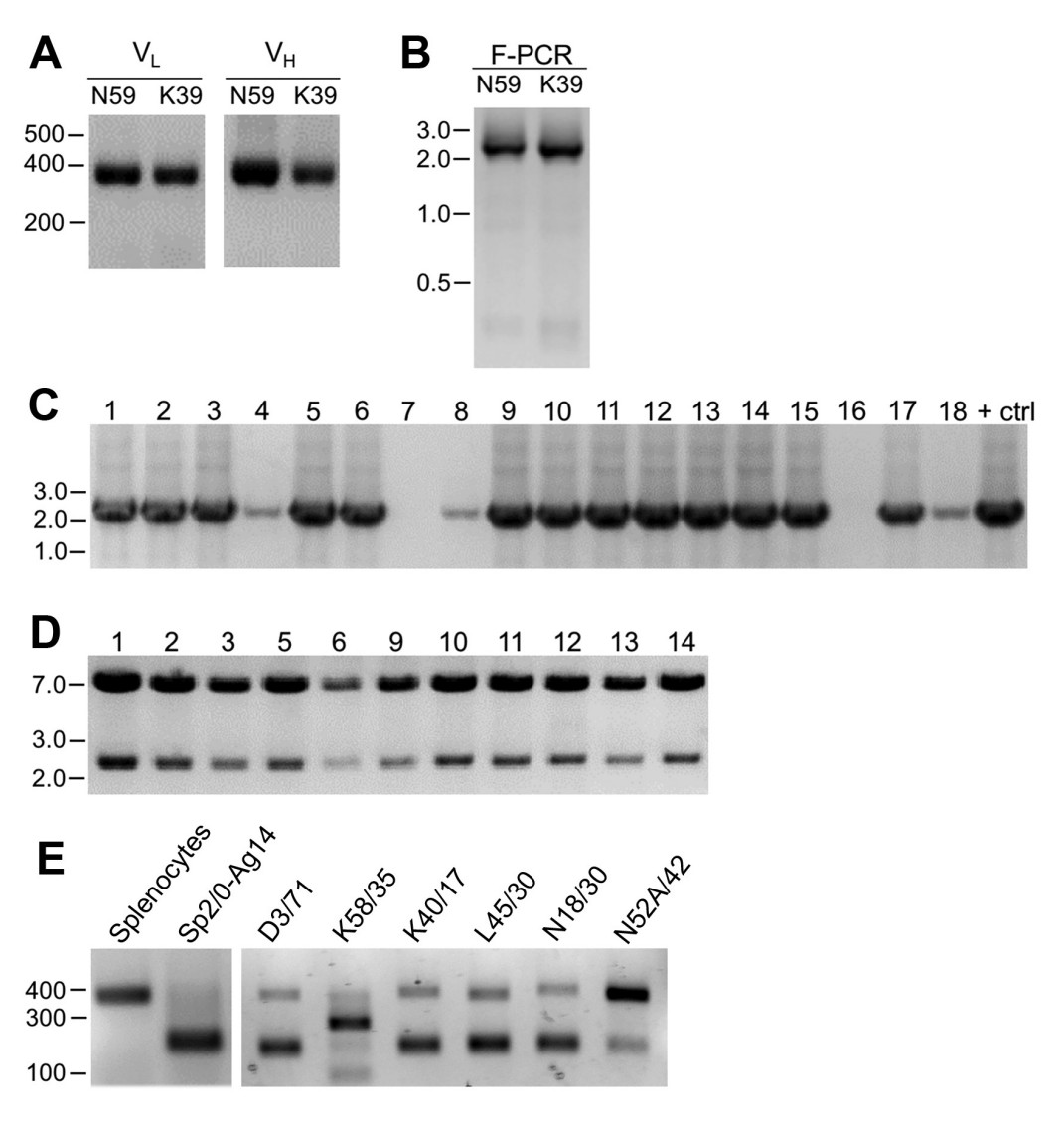

**Figure 2.** Cloning of $V_L$ and $V_H$ domain sequences from hybridomas into the R-mAb expression plasmid. (**A**) Agarose gel analysis of $V_L$ and $V_H$ domain PCR products amplified from cDNA synthesized from RNA extracted from the N59/36 (anti-NR2B/GRIN2B) and K39/25 (anti-Kv2.1/KCNB1) hybridomas. The expected size of mouse IgG $V_L$ and $V_H$ domains is ≈360 bp. (**B**) Agarose gel analysis of $V_H$ and digested $V_L$ fragments joined by fusion PCR (F-PCR) to the P1316-derived joining fragment to create a dual IgG chain cassette. (**C**) Agarose gel analysis of colony PCR samples of transformants from the N59/36 R-mAb project. (**D**) Agarose gel analysis of products of restriction enzyme digestion of N59/36 plasmid DNA with NotI and AscI. The plasmid backbone is seven kbp, and the intact insert comprising the $V_L$ and $V_H$ domains and the intervening joining fragment is 2.4 kbp. (**E**) Agarose gel analysis of PCR products of $V_L$ domain cDNA synthesized from RNA extracted from mouse splenocytes, the fusion partner Sp2/0-Ag14, and various hybridomas after digestion with the BciVI restriction enzyme to cleave the Sp2/0-Ag14-derived aberrant light chain product. The intact $V_L$ domains are ≈360 bp, and the digested aberrant light chains ≈180 bp.

DOI: https://doi.org/10.7554/eLife.43322.004

expression in mammalian cells, and R-mAb validation and sequencing as detailed in the subsequent sections.

While a number of *bona fide* R-mAbs were isolated using this approach, we found a high degree of variability in the number of colony PCR- and restriction enzyme digest- verified positive plasmids that yielded functional expression. A major obstacle in cloning functionally rearranged IgG

sequences from many mouse hybridomas is the presence of an aberrant kappa IgG light chain transcript expressed by the Sp2/0-Ag14 (Sp2/0) hybridoma (*Carroll et al., 1988*) that frequently serves as a 'myeloma' partner for fusion with mouse splenocytes to generate mAb-producing hybridomas (*Shulman et al., 1978*), and that was used as the fusion partner in all of our mAb generation efforts (*Bekele-Arcuri et al., 1996*; *Gong et al., 2016*). The source of this non-productive IgG light chain is the MOPC-21 myeloma cell line used to generate the Sp2/0 hybridoma (*Shulman et al., 1978*). As we experienced, and as previously reported by others (*Carroll et al., 1988*), aberrant chain mRNA expression varies greatly among distinct hybridoma cell lines but in certain cases can exceed the levels of functional light chain transcripts. For certain of our projects, this resulted in >90% of the colony PCR positive clones failing to produce detectable levels of functional R-mAbs, thus necessitating a high volume of screening.

As such, we sought to eliminate this aberrant light chain during the cloning process. We treated the $V_L$ PCR products with the restriction enzyme BciVI. The restriction site for this enzyme is present in the $V_L$ region of the aberrant Sp2/0-derived transcript, but is predicted to occur at a low frequency in functional mouse $V_L$ kappa sequences (*Juste et al., 2006*). We used $V_L$ PCR products derived from the Sp2/0 cell line and from pooled BALB/c mouse splenocytes as positive and negative controls, respectively, for sensitivity to BciVI digestion. Due to the exclusive presence of aberrant light chain in Sp2/0 cells, $V_L$ PCR products from these cells were completely digested, as shown by the decreased size of the $V_L$ PCR products from ≈360 bp typical of $V_L$ PCR products (see *Figure 2A* for examples) to the ≈180 bp fragment that results from BciVI digestion (*Figure 2E*). In contrast, the sample prepared from the pooled mouse splenocytes was not detectably affected by BciVI digestion (*Figure 2E*). Treatment of $V_L$ PCR products from various Sp2/0-derived hybridomas with BciVI resulted in varying degrees of digestion, yielding different proportions of the bands representing the intact $V_L$ PCR product of ≈360 bp and the cleaved aberrant SP2/0-derived $V_L$ fragment of ≈180 bp (*Figure 2E*). After BciVI digestion was incorporated into the protocol, DNA sequencing of 149 colony PCR-positive clones from 26 different hybridomas revealed that only 12 (8%) still contained the aberrant $V_k$ light chain (*Table 1*). In certain cases, digestion of hybridoma $V_L$ PCR products resulted in fragments of unexpected sizes (see the K58/35 lane in *Figure 2E*), indicating, as predicted by an earlier bioinformatics analysis (*Juste et al., 2006*), that in rare cases (in our hands, 3/248 clones pursued to this step) the BciVI restriction site was also present in these functionally rearranged splenocyte-derived $V_L$ genes. As such, we attempted to clone these $V_L$ PCR products without BciVI treatment. As one example, the splenocyte-derived $V_k$ light chain PCR products from the K58/35 hybridoma were sensitive to BciVI digestion, which necessitated their cloning in the absence of the BciVI digestion step. This project yielded somewhat lower frequency of clones that produced functional R-mAbs able to detect target antigen (≈37%) than, on average, those containing splenocyte-derived $V_L$ PCR products refractory to BciVI digestion (≈48%; *Supplementary file 1*). For the bulk of mAbs encoded by splenocyte-derived $V_L$ PCR products resistant to BciVI digestion, ligations were performed following BciVI digestion, and 10–14 candidate clones that were colony PCR-positive were selected. Plasmid DNA was digested with AscI/NotI restriction enzymes to confirm the correct insert (*Figure 2D*). On average, ≈93% of all clones subjected to restriction analysis passed this screening step.

We next transfected the colony PCR and restriction digest-validated plasmids into COS-1 cells and after a 3 to 6 day incubation, tested the conditioned culture media for production of target-specific R-mAbs. While the parent mAbs had previously been validated for efficacy and specificity in a variety of applications (IF-ICC, IB and IHC on brain samples), we selected the IF-ICC assay in transiently transfected heterologous cells for R-mAb validation. This method was chosen because it is high-throughput, employing 96 well microtiter plates, it requires only a small amount of R-mAb sample, and the robust difference between target-expressing and non-expressing cells in the same sample allows for sensitivity and clarity of results. Importantly, each of the parent mAbs had been previously validated in this procedure. We took advantage of the fact that in most cases we switched the IgG subclass during conversion of the hybridoma-derived mAb into the corresponding R-mAb, allowing for their separate detection by subclass-specific secondary Abs (*Manning et al., 2012*). This assay involved expressing the full-length target protein in transiently transfected COS-1 cells cultured in individual wells of a black polystyrene 96 well clear bottom plate that allows for microscopic visual analysis and imaging using indirect immunofluorescence. For each of 1 to 15 R-mAb candidate clones to be assayed from a given project, a set of replicate wells were prepared expressing the

**Table 1.** Aberrant $V_L$ sequences remaining after BciVI digestion.

Table delineates for specific mAb cloning projects the total number of clones sequenced, and the number of sequenced clones with $V_L$ chains corresponding to the Sp2/0-Ag14 hybridoma-derived aberrant $V_L$ transcript.

| mAb | Number of clones sequenced | Number of clones with aberrant chain | % with aberrant chain |
|---|---|---|---|
| K7/45 | 5 | 0 | 0 |
| K9/40 | 7 | 2 | 28.6 |
| K13/31 | 10 | 0 | 0 |
| K14/16.2 | 2 | 0 | 0 |
| K14/16.2.1 | 9 | 2 | 22.2 |
| K17/70 | 6 | 1 | 16.7 |
| K28/86 | 3 | 0 | 0 |
| K36/15 | 6 | 1 | 16.7 |
| K74/71 | 6 | 0 | 0 |
| K75/41 | 3 | 0 | 0 |
| K87A/10 | 3 | 0 | 0 |
| L6/60 | 3 | 0 | 0 |
| L21/32 | 4 | 0 | 0 |
| L23/27 | 6 | 0 | 0 |
| L61/14 | 7 | 1 | 14.3 |
| N59/20 | 2 | 2 | 100 |
| N70/28 | 1 | 0 | 0 |
| N86/8 | 8 | 2 | 25.0 |
| N86/38 | 22 | 0 | 0 |
| N100/13 | 6 | 0 | 0 |
| N103/31 | 4 | 1 | 25.0 |
| N103/39 | 5 | 0 | 0 |
| N105/13 | 5 | 0 | 0 |
| N106/36 | 6 | 0 | 0 |
| N116/14 | 3 | 0 | 0 |
| N297/59 | 7 | 0 | 0 |
| Total | 149 | 12 | 8.1 |

DOI: https://doi.org/10.7554/eLife.43322.005

given target protein. After fixation and permeabilization, the individual wells were then immunolabeled with either the hybridoma-generated mAb alone, the R-mAb alone, or the mAb and R-mAb together. Each well was subsequently incubated with a cocktail of the two distinct subclass-specific secondary Abs, one specific for the respective mAb and one specific for the subclass-switched R-mAb mouse IgG subclasses, and both conjugated to spectrally distinct Alexa Fluors. The 'mAb only' well was used to demonstrate that the target protein was expressed in a subset of the transiently transfected cells, and that the only detectable secondary Ab labeling was for the IgG subclass of the parent hybridoma-generated mAb. Similarly, the 'R-mAb only' wells were used to show that the R-mAb labeled a comparable number of cells, and that the only detectable secondary Ab labeling was for the IgG subclass of the subclass-switched R-mAb. The wells containing both the parent mAb and candidate R-mAb were used to show that the mAb and R-mAb gave indistinguishable labeling patterns at both the cellular and subcellular level and could be detected separately using subclass-specific secondary Abs. Numerous examples of this assay are shown in the following sections focusing on specific R-mAb projects. We note that only in rare cases was the labeling for one or both secondaries noticeably depressed in the well containing both mAb and R-mAb relative to the wells with these primary antibodies alone, as would occur due to competitive binding to the

same epitope. However, this was generally not apparent, suggesting that in this assay system, in which the target protein was overexpressed, we were typically operating under conditions of antigen excess.

In comparing the overall results from 180 recent projects that had BciVI-resistant splenocyte-derived $V_L$ PCR products, and that were taken through this entire pipeline one time (*Supplementary file 1*), we found that a range of 1–15 colony PCR- and restriction digest-positive candidate R-mAbs per project were evaluated in the COS-ICC assay (919 total, mean/project = 5.11 ± 0.23 S.E.M.). For these 180 projects, cloning was performed after BciVI digestion, and all restriction digest-validated candidates were tested for functional mAb production. Of these 180 projects, 72% (129 projects) yielded at least one positive R-mAb on their first pass through the pipeline. A retrospective analysis of these 129 successful projects revealed a range of 1 to 13 restriction digest-validated candidates were evaluated (mean/project = 5.74 ± 0.26 S.E.M.), with an overall success rate of 48.4% for all 741 restriction digest-validated candidate R-mAbs tested. A parallel analysis of the 51 projects that did not yield a positive R-mAb on their first pass through the pipeline revealed a similar range of 1 to 15 restriction digest-validated candidates per project evaluated. However, an overall lower number of colony PCR- and restriction digest-positive candidates were evaluated in the COS-ICC assay (178 total, mean/project = 3.5 ± 0.37 S.E.M.) in these projects than for the successful projects. A statistically significant difference (two-tailed P value = $2.48 \times 10^{-6}$) existed between the number of candidates evaluated in successful versus unsuccessful projects. The per-project COS-ICC rate success was impacted by number of clones tested. For 1–3 clones tested (66 projects), the project success rate was 50%, for 4–6 (62 projects) it was ≈ 77%, and for ≥7 (52 projects) it was ≈92%. However, the per clone success rate was similar between the three bins (≈ 45% vs. ≈ 39% vs. ≈ 45%).

Following validation in the COS-ICC assay, a subset of positive clones for each project was subjected to DNA sequencing. We employed a set of sequencing primers that allowed for determination of sense and antisense strands of the $V_H$ and $V_L$ domain-encoding cDNA inserts that were unique to each R-mAb. The sequences were searched against the NCBI database, and against a custom database that contained the $V_H$ and $V_L$ domain sequences of the parent P1316 plasmid, the $V_L$ domain of the aberrant Sp2/0 cell line, and the $V_H$ and $V_L$ domain sequences of all of the R-mAbs we had cloned to date. The set of COS-ICC validated R-mAb plasmids derived from a single hybridoma that had matching sequences unique from any sequences in the custom database were subsequently archived as frozen plasmid DNA and bacterial glycerol stocks, and their sequence used as the archival sequence of that particular R-mAb.

## Effective cloning and IgG subclass switching of a widely used monoclonal antibody

We initiated our R-mAb cloning efforts with the K28/43 mAb, a mouse mAb specific for the neural scaffolding protein PSD-95. This mAb is widely used as a marker of excitatory synapses (*e.g.*, as of 1/1/19 ≈520 publications have cited the use of the K28/43 mAb as obtained from the UC Davis/NIH NeuroMab Facility alone). While the K28/43 mAb is already of a less common IgG subclass (IgG2a), generating a subclass switched version with an alternate IgG subclass would provide greater flexibility in its use in multiplex labeling experiments, and also provide proof of concept that we could use our process to effectively generate subclass switched R-mAbs with efficacy and specificity comparable to the parent mAb. We generated $V_L$ and $V_H$ domain cDNA fragments from the cryopreserved K28/43 hybridoma and cloned them into the original P1316 plasmid that contains a mouse IgG1 $C_H$ domain (*Crosnier et al., 2010*). COS-1 cells were transiently transfected with the resultant plasmids. We evaluated the conditioned medium from the transfected cells for the presence of the subclass-switched K28/43R IgG1 R-mAb by ICC against fixed and permeabilized transiently transfected COS-1 cells expressing full-length PSD-95 (*Figure 3A*). We note that we have designated the recombinant R-mAb versions of each of our mAbs by a capital 'R' after the clone designation, in this case the R-mAb cloned from the K28/43 hybridoma is designated K28/43R. We screened the COS-1 cell-generated R-mAbs for functional R-mAb immunoreactivity in the 96-well IF-ICC assay detailed in the previous section. After primary antibody labeling, all wells received both IgG1- and IgG2a-subclass-specific, fluorescently labeled secondary antibodies. As shown in *Figure 3A*, as expected, the sample receiving only the native K28/43 hybridoma-generated mAb exhibited a signal corresponding to the IgG2a subclass-specific secondary Ab (red) with no detectable signal for the IgG1 subclass-

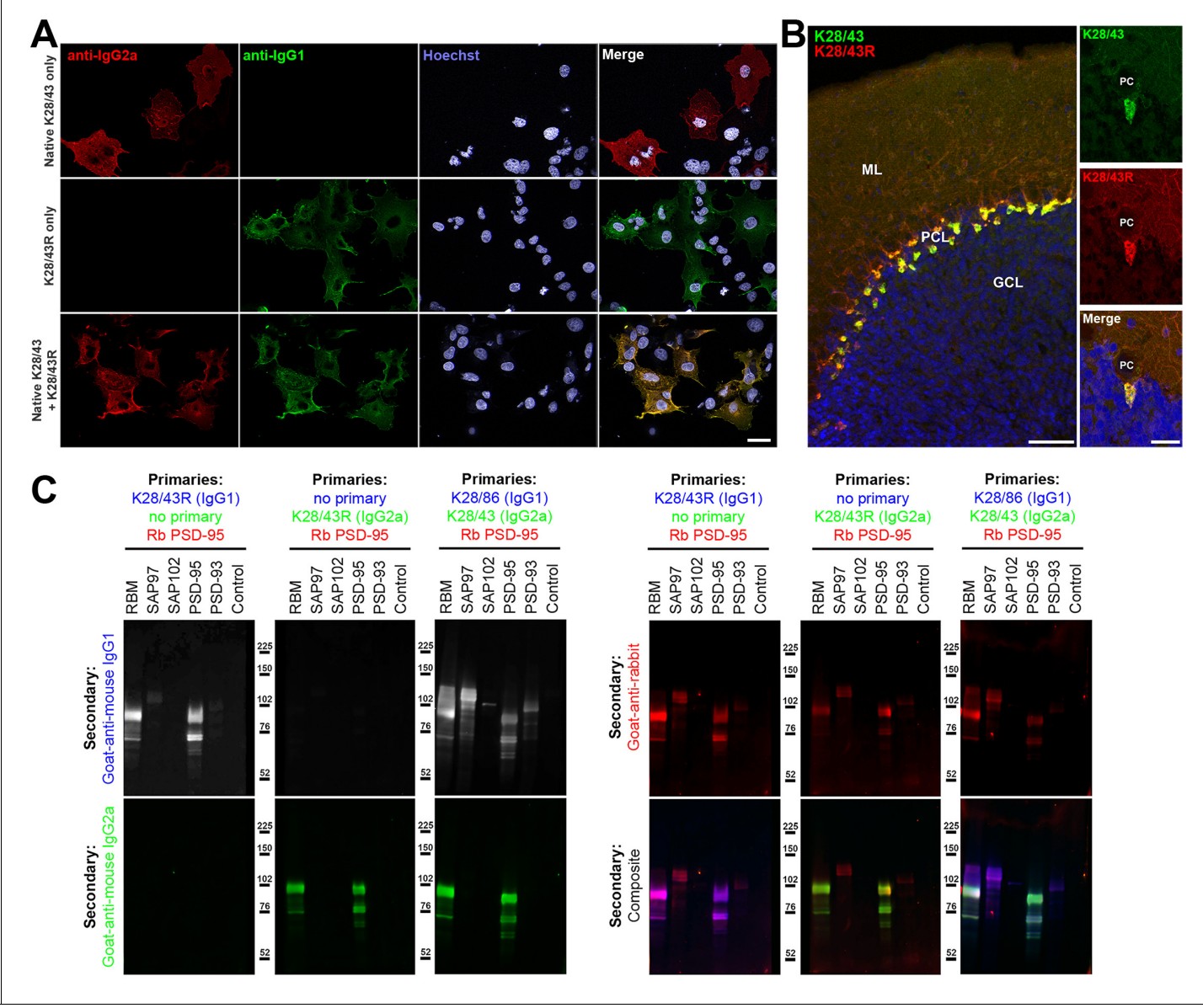

**Figure 3.** Validation of subclass-switched anti-PSD-95 K28/43R R-mAb. (**A**) Validation of the K28/43R R-mAb in heterologous cells. COS-1 cells transiently transfected to express human PSD-95 in a subset of cells were immunolabeled with K28/43 mAb (IgG2a) alone (top row), K28/43R R-mAb (IgG1) alone (middle row), or K28/43 mAb plus K28/43R R-mAb (bottom row). Immunolabeling in all samples was detected with a cocktail of anti-mouse IgG2a (red, for the K28/43 mAb) and anti-mouse IgG1 (green, for the K28/43R R-mAb) subclass-specific Alexa Fluor conjugated secondary antibodies. Labeling in blue is for the DNA-specific dye Hoechst 33258 and shows nuclei of both transfected and untransfected cells. Scale bar in the lower right merged panel = 30 µm and holds for all panels in A. (**B**) Validation of the K28/43R R-mAb in brain sections. A brain section from an adult rat was immunolabeled with K28/43 mAb plus K28/43R R-mAb and immunolabeling detected with a cocktail of anti-mouse IgG2a (red, for K28/43 mAb) and anti-mouse IgG1 (green, for K28/43R R-mAb) subclass-specific Alexa Fluor conjugated secondary antibodies. Cell nuclei are labeled with the DNA-specific dye Hoechst 33258 (blue). The region of interest shown is from cerebellar cortex. Scale bar in the left panel = 100 µm, and in the right merged panel = 30 µm. (**C**) Immunoblots against brain membranes and COS cell lysates over-expressing various members of the MAGUK superfamily of scaffolding proteins. To confirm expression of MAGUK proteins, immunoblots were probed with rabbit anti-PSD-95 (red). K28/86 is an anti-MAGUK mAb. Primary antibodies were detected with the appropriate combinations of fluorescently labeled species-specific anti-rabbit and subclass-specific anti-mouse IgG secondary Abs as indicated. Control indicates COS cells transfected with an empty vector.

DOI: https://doi.org/10.7554/eLife.43322.006

specific secondary Ab (green). Conversely, labeling with the K28/43R R-mAb alone produced only an IgG1 subclass-specific green signal demonstrating a successful IgG subclass switch for the R-mAb (*Figure 3A*). Simultaneous multiplex labeling with the hybridoma-generated mAb and a positive R-mAb resulted in an identical pattern of immunolabeling at the cellular level in the specific cells recognized, and at the subcellular level as to the pattern of immunolabeling within the labeled cells, as shown by the uniform hue of the signal in the merged panel, indicating that both the mAb and R-mAb were recognizing the same target (*Figure 3A*). We next performed multiplex immunofluorescent IHC on adult mouse brain sections with the K28/43 mAb and the K28/43R R-mAb. As shown in *Figure 3B*, as detected with secondary Abs specific for their respective mouse IgG subclasses, the signal from these two primary antibodies was indistinguishable in its laminar and subcellular pattern in cerebellar cortex, consistent with previous studies of PSD-95 (*e.g.*, *Kistner et al., 1993*). Both signals were especially intense in the terminal pinceau of basket cells located adjacent to the Purkinje cell layer, and both signals were also present in the molecular layer, and for the most part lacking in the granule cell layer (*Figure 3B*). This demonstrates that consistent with the validation in heterologous COS-1 cells expressing exogenous PSD-95, the K28/43 R-mAb can be used reliably for multiplex immunolabeling of endogenous PSD-95 in brain sections.

The specificity of recombinant K28/43R was also assessed on immunoblots of samples from COS-1 cells exogenously expressing various representatives of the MAGUK superfamily of scaffolding proteins, of which PSD-95 is one member (*Figure 3C*). Immunoblots were probed with a rabbit polyclonal antibody raised against PSD-95 and that also cross-reacts with SAP97 as a positive control, and recombinant K28/43R in either the IgG1 or IgG2a mouse IgG subclass form (*Figure 3C*). Generation of the IgG2a expression plasmid is described below. Both the IgG1 and IgG2a subclass isoforms of the K28/43R R-mAb gave identical immunolabeling patterns against samples from rat brain and COS-1 cells overexpressing PSD-95. Moreover, the pattern of R-mAb immunolabeling was similar to that obtained with the rabbit polyclonal antibody, and absent against samples of COS-1 cells exogenously expressing other MAGUK superfamily members (*Figure 3C*). To confirm expression of these MAGUK proteins, immunoblots were probed with rabbit polyclonal anti-PSD-95 and with the hybridoma-generated K28/43 mAb (IgG2a) or a mAb that recognizes all mammalian MAGUK proteins mAb (K28/86; IgG1) (*Rasband et al., 2002*) (*Figure 3C*). These initial results demonstrated that we could use this cloning and expression process to generate and validate subclass-switched R-mAbs that recapitulate the immunolabeling characteristics of the native mAbs.

### Generation of an R-mAb expression plasmid with an IgG2a C$_H$ backbone allows for effective subclass switching of mAbs from prevalent IgG1 subclass

Most ($\approx$70%) mouse IgG mAbs are of the IgG1 subclass (*Manning et al., 2012*). Generating R-mAbs employing the efficient V$_H$ and V$_L$ cloning approach developed by Gavin Wright and colleagues includes their subsequent insertion into an expression plasmid that yields R-mAbs of the mouse IgG1 subclass due to the presence of the mouse $\gamma$1 C$_H$ domain in the plasmid backbone (*Crosnier et al., 2010*). To generate R-mAbs of less common mouse IgG subclasses, we modified this plasmid by replacing the $\gamma$1 C$_H$ domain with a mouse $\gamma$2a C$_H$ domain. We amplified the $\gamma$2a C$_H$ domain sequence from cDNA generated from the hybridomas producing the K28/43 IgG2a mAb and then replaced the $\gamma$1 C$_H$ domain present in the K28/43R plasmid with this $\gamma$2a C$_H$ domain using standard cloning. This plasmid was sequenced verified and validated for expression of a K28/43R R-mAb of the IgG2a subclass (*Figure 3C*).

We subsequently used this plasmid as the target cloning vector for generation and expression of numerous R-mAb plasmids encoding functional IgG2a R-mAbs. Our strategy entailed cloning all native IgG1 and IgG2b mAbs into this plasmid to generate forms distinct from the parent native mAbs. Toward this end, we have successfully cloned and validated as functional R-mAbs a total of 178 mAbs (*Supplementary file 2*). This includes 148 mAbs of distinct subclasses converted to IgG2a R-mAbs (125 IgG1, 21 IgG2b, and 3 IgG3), and 29 IgG2a mAbs that were retained in their native IgG subclass (*Supplementary file 2*). We also converted the IgG2a mAb K28/43 to an IgG1 R-mAb. Each of these R-mAbs have been assigned unique RRID numbers in the Antibody Registry (*Supplementary file 2*) and all will be deposited in plasmid form at Addgene, a subset of which are already available (https://www.addgene.org/James_Trimmer/).

## Multiplex brain immunofluorescent labeling with subclass switched R-mAbs

One benefit of subclass switching R-mAbs is the ability to perform multiplex immunolabeling not previously possible due to IgG subclass conflicts. Examples of such enrichment in cellular protein localization are shown in *Figure 4*. *Figure 4A* shows labeling with the subclass switched IgG2a R-mAb derived from the widely used pan-voltage-gated sodium channel or 'pan-Nav channel' IgG1 mAb K58/35 (*Rasband et al., 1999*). Like the corresponding mAb, K58/35 R-mAb gives robust labeling of Nav channels concentrated on the axon initial segment (AIS, arrows in *Figure 4A* main panel), and at nodes of Ranvier (arrows in *Figure 4A* insets). Importantly, subclass switching allowed Nav channel labeling at nodes to be verified by co-labeling with K65/35 an IgG1 subclass antibody directed against CASPR, a protein concentrated at paranodes (*Menegoz et al., 1997*; *Peles et al., 1997*). Similarly, simultaneous labeling for the highly-related GABA-A receptor β1 and β3 subunits (*Zhang et al., 1991*) with their widely used respective mAbs N96/55 and N87/25 could not be performed as both are IgG1 subclass. Switching the N96/55 mAb to the IgG2a N96/55R R-mAb allowed simultaneous detection of these two highly-related but distinct GABA-A receptor subunits. In *Figure 4B* localization of GABA-A receptor β1 and β3 subunits appeared completely non-overlapping in separate layers of cerebellum. *Figure 4C* illustrates localization of protein Kv2.1 and AnkyrinG in separate subcellular neuronal compartments. Labeling for the K89/34R R-mAb (IgG2a, red), specific for the Kv2.1 channel, highly expressed in the plasma membrane of the cell body and proximal dendrites (arrows in panel C1) is shown together with labeling for N106/65 (green), an IgG1 mAb specific for AnkyrinG, a scaffolding protein highly expressed in the AIS (arrows in panel C2) and at nodes of Ranvier. Subclass switching N229A/32 (IgG1, GABA-AR α6) to IgG2a, allows comparison with Kv4.2 potassium channel (K57/1, IgG1) in the cerebellum where both are highly expressed in the granule cell layer (*Figure 4D*). While both are prominently found in the glomerular synapses present on the dendrites of these cells, simultaneous labeling reveals that some cells express both (*Figure 4D*, magenta) while others appear to predominantly express Kv4.2 (*Figure 4D*, blue). Labeling for both proteins is in contrast to that for mAb N147/6 (IgG2b) which recognizes all isoforms of the QKI transcription factor and labels oligodendrocytes within the granule cell layer and throughout the Purkinje cell layer (PCL, green). In *Figure 4E*, localization of pan-QKI (N147/6, IgG2b, blue) is compared with GFAP (N206A/8, IgG1, green) predominantly thought to be in astrocytes. Surprisingly many (but not all) cells co-label both proteins. We also labeled cortical neurons with these two mAbs (pan-QKI in blue, GFAP in green). Multiplex labeling for the neuron-specific Kv2.1 channel, using subclass-switched K89/34R (IgG2a, red) confirms non-neuronal localization of both proteins. Lastly, we labeled for the postsynaptic scaffold protein PSD-93 using R-mAb N18/30R (IgG2a, red), which in the cerebellum is prominently localized to Purkinje cell somata and dendrites (*Brenman et al., 1998*). As shown in *Figure 4F*, the R-mAb labeling is consistent with the established localization of PSD-93. Because of subclass switching the N18/30R R-mAb, this labeling can now be contrasted with labeling for the excitatory presynaptic terminal marker VGluT1, labeled with mAb N28/9 (IgG1, blue), which exhibits robust labeling of parallel fiber synapses in the molecular layer, and glomerular synapses in the granule cell layer. Together these results demonstrate the utility of employing subclass-switched R-mAbs to obtain labeling combinations not possible with native mAbs.

## Recovery of functional R-mAbs from a non-viable hybridoma

While not a common event, mAb-producing B cell hybridomas can lose their mAb production due to genetic instability (*Morrison and Scharff, 1981*; *Frame and Hu, 1990*; *Castillo et al., 1994*), mycoplasma contamination leading to amino acid depletion and cytopathic effects (*Drexler and Uphoff, 2002*), or other factors. Non-optimal cryopreservation due to problems during the freezing process itself or inadequate storage conditions can result in non-recoverable frozen seeds. As the method described here for cloning of $V_H$ and $V_L$ domain sequences from cryopreserved hybridomas does not require expansion of cells in culture prior to RNA extraction, we speculated that it may be possible to use this approach to generate functional R-mAbs from even those hybridoma cell lines that are no longer viable in cell culture.

We had in our cryopreserved hybridoma archive a hybridoma cell line that produced the D3/71 mAb. This hybridoma cell line had been generated in 1994 among a series of projects targeting the

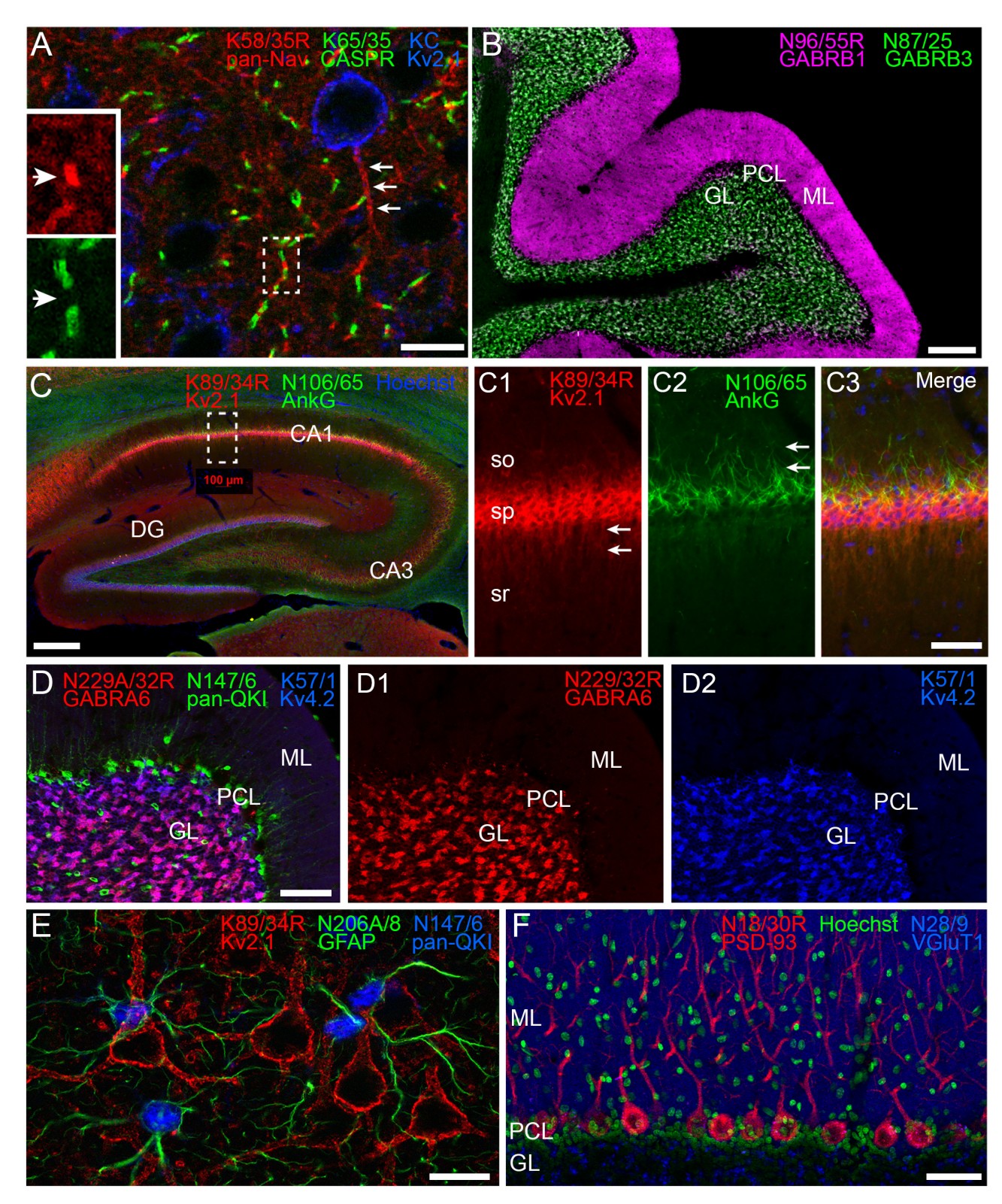

**Figure 4.** Multiplex immunolabeling with subclass-switched recombinant antibodies in adult rat brain. (**A**) A section from neocortex labeled with anti-pan-Nav R-mAb K58/35R (IgG2a, red) at nodes of Ranvier and AIS (arrows), anti-CASPR mAb K65/35 (IgG1, green) at paranodes, and anti-Kv2.1 rabbit polyclonal (KC) antibody (blue) on somata and proximal dendrites. Scale bar = 150 µm. Insets (dashed box) show details of labeling for pan-Nav (red) and CASPR (green) at the node and paranodes (arrows), respectively, at a single node of Ranvier as indicated by box in main panel. (**B**) A section

*Figure 4 continued on next page*

*Figure 4 continued*

through cerebellum showing labeling with anti-GABA-AR β1 R-mAb N96/55R (IgG2a, magenta) in the molecular layer (ML), and anti-GABA-AR β3 mAb N87/25 (IgG1, green) in the granule cell layer (GL). PCL = Purkinje cell layer. Scale bar = 150 μm. (C) A section through hippocampus labeled with anti-Kv2.1 R-mAb 89/34R (IgG2a, red) on somata and proximal dendrites, anti-AnkyrinG mAb N106/65 (IgG2b, green) on AIS, and nuclear stain Hoechst 33258 (blue). Scale bar = 150 μm. Panels C1-C3 show magnified details of labeling for pan-Kv2.1 (red) on somata and proximal dendrites (arrows in C1), and anti-AnkyrinG (green) on AIS (arrows in C2). Scale bar = 50 μm (C1–C3). (D) A section through cerebellum labelled with anti-GABA-AR α6 R-mAb K229A/32R (IgG2a, red) in the granule cell layer (GL), anti-pan-QKI mAb N147/6 (IgG2b, green) labeling glial cells in/near the Purkinje cell layer (PCL), and anti-Kv4.2 mAb K57/1 (IgG1, blue) labeling the granule cell layer (GL). Scale bar = 30 μm. (E) A section from neocortex labelled with anti-Kv2.1 R-mAb 89/34R (IgG2a, red) on somata and proximal dendrites of neurons, and anti-GFAP mAb N206A/8 (IgG1, green) and anti-pan-QKI mAb N147/6 (IgG2b, blue) labeling glial cell processes and cell bodies respectively. Scale bar = 15 μm. (F) A section through cerebellum showing labeling with anti-PSD-93 R-mAb N18/30R (IgG2a, red) in the cell bodies and dendrites of Purkinje cells, the nuclear stain Hoechst 33258 (green) and anti-VGluT1 mAb N28/9 (IgG1, blue) in the molecular layer (ML). PCL = Purkinje cell layer. Scale bar = 10 μm.

DOI: https://doi.org/10.7554/eLife.43322.007

Kv2.1 voltage-gated potassium channel α subunit, projects that also yielded the D4/11 mAb (*Bekele-Arcuri et al., 1996*). The D3/71 mAb has particular value in binding at a site distinct from other anti-Kv2.1 mAbs, which unlike most other available anti-Kv2.1 mAbs (and pAbs), remains intact in truncated isoforms of Kv2.1 found in patients with severe neurodevelopmental disorders linked to de novo frameshift or nonsense mutations in the KCNB1 gene (*de Kovel et al., 2016*; *Marini et al., 2017*). Unlike our experience with other cryopreserved hybridomas in our collection, when we attempted to resuscitate this cryopreserved hybridoma cell line over 20 years after it was cryopreserved, the D3/71 hybridoma cells were no longer viable. We extracted RNA from a cryopreserved vial of D3/71 hybridoma cells in an attempt to generate a functional D3/71R R-mAb. Cloning was performed as described above including amplification of D3/71 $V_H$ and $V_L$ domain cDNAs (*Figure 5A–B*) and their insertion into the mouse IgG2a expression plasmid. Most of the bacterial colonies tested (20/24) were positive for the insert by PCR (*Figure 5C*), and subsequent restriction

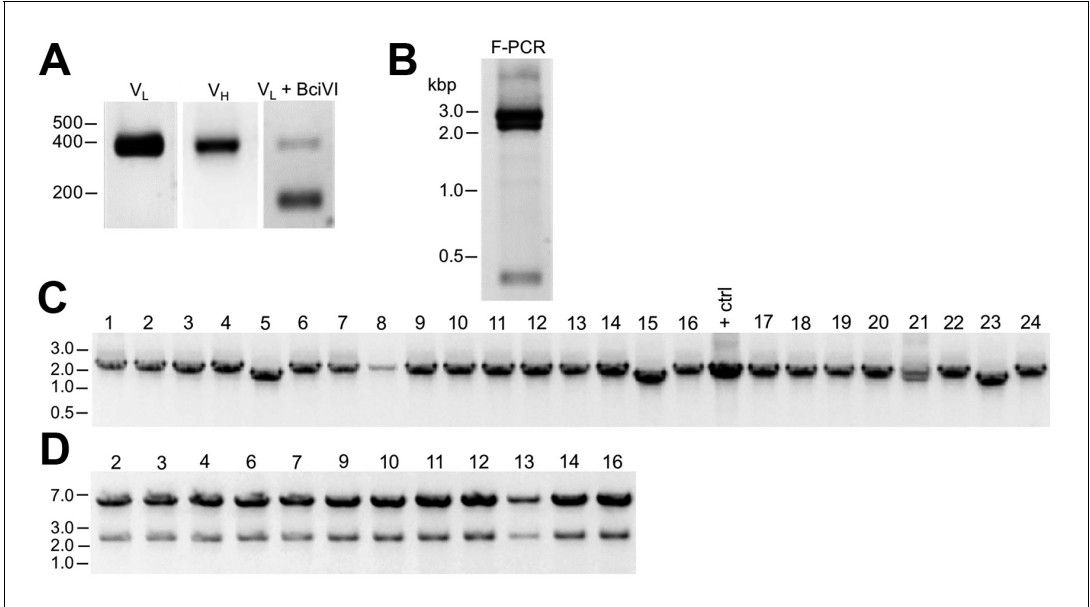

**Figure 5.** Cloning of anti-Kv2.1 D3/71 $V_L$ and $V_H$ domain cDNAs from a nonviable hybridoma. (A) Agarose gel analysis of PCR amplified $V_L$ and $V_H$ domains from cDNA synthesized from RNA extracted from the non-viable D3/71 hybridoma. The panel to the right shows the $V_L$ after digestion with the BciVI restriction enzyme to cleave the Sp2/0-Ag14-derived aberrant light chain product. The expected size of mouse IgG $V_L$ and $V_H$ domains is ≈360 bp, and of the cleaved aberrant $V_L$ domain is ≈180 bp. (B) Agarose gel analysis of D3/71 $V_H$ and digested $V_L$ fragments joined by fusion PCR (F-PCR) to the P1316 joining fragment to create a dual IgG chain cassette. (C) Agarose gel analysis of colony PCR samples of transformants from the of D3/71 R-mAb project. (D) Agarose gel analysis of products of restriction enzyme digestion of D3/71 plasmid DNA with NotI and AscI. The plasmid backbone is seven kbp, and the intact insert comprising the $V_L$ and $V_H$ domains and the intervening joining fragment is 2.4 kbp.

DOI: https://doi.org/10.7554/eLife.43322.008

analysis of 12 of these positive clones confirmed that they each contained the full-length $V_L$-joining fragment-$V_H$ cassette (*Figure 5D*). We next tested whether these plasmids encoded functional D3/71R R-mAb in the COS-1-ICC assay, using a distinct native anti-Kv2.1 IgG1 mAb K89/34 (*Misonou et al., 2004*) and the subclass-switched K89/34R R-mAb (*Figure 6B*) as controls (*Mandikian et al., 2014*; *Bishop et al., 2015*). A subset of the recovered and subclass-switched D3/71R R-mAbs were positive in this assay (*Figure 6A*). Moreover, the D3/71R R-mAb exhibits immuno-labeling of endogenous Kv2.1 in neurons in mouse neocortex that recapitulates the previously

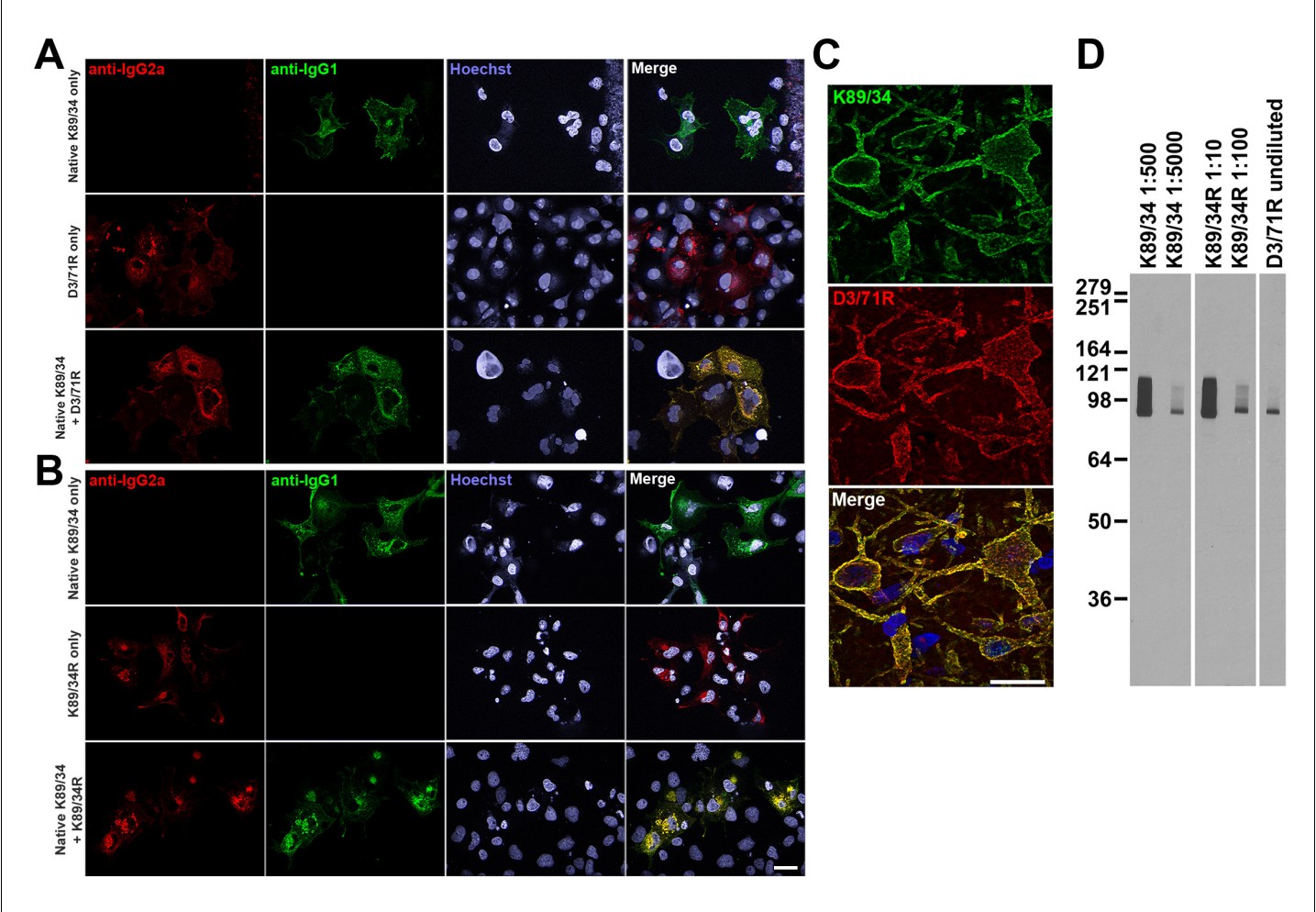

**Figure 6.** Recovery of a functional anti-Kv2.1 D3/71R R-mAb from nonviable hybridomas. (**A**) Validation of the D3/71R R-mAb in heterologous cells. COS-1 cells transiently transfected to express rat Kv2.1 in a subset of cells were immunolabeled with K89/34 mAb (IgG1) alone (top row), D3/71R R-mAb (IgG2a) alone (middle row), or K89/34 mAb plus D3/71R R-mAb (bottom row). Immunolabeling in all samples was detected with a cocktail of anti-mouse IgG1 (green, for the K89/34 mAb) and anti-mouse IgG2a (red, for the D3/71R R-mAb) subclass-specific Alexa Fluor conjugated secondary antibodies. (**B**) Validation of the subclass-switched K89/34R R-mAb in heterologous cells. COS-1 cells transiently transfected to express rat Kv2.1 in a subset of cells were immunolabeled with K89/34 mAb (IgG1) alone (top row), K89/34R R-mAb (IgG2a) alone (middle row), or K89/34 mAb plus K89/34R R-mAb (bottom row). Immunolabeling in all samples was detected with a cocktail of anti-mouse IgG1 (green, for the K89/34 mAb) and anti-mouse IgG2a (red, for the K89/34R R-mAb) subclass-specific Alexa Fluor conjugated secondary antibodies. Labeling in blue in panels A and B is for the DNA-specific dye Hoechst 33258 and shows nuclei of both transfected and untransfected cells. Scale bar in the lower right merged panel = 30 μm and holds for all panels in A and B. (**C**) Validation of the D3/71R R-mAb in brain sections. A brain section from an adult rat was immunolabeled with K89/34 mAb plus D3/71 R-mAb and the immunolabeling detected with a cocktail of anti-mouse IgG1 (green, for the K89/34 mAb) and anti-mouse IgG2a (red, for the D3/71R R-mAb) subclass-specific Alexa Fluor conjugated secondary antibodies. Cell nuclei are labeled with the DNA-specific dye Hoechst 33258 (blue). Region of interest shown is from neocortex. Scale bar = 30 μm. (**D**) Strip immunoblots on a crude rat brain membrane fraction immunolabeled with the K89/34 mAb, the K89/34R R-mAb, and the D3/71 R-mAb as indicated. Immunolabeling was detected on autoradiography film after treatment of strip immunoblots with HRP-conjugated anti-mouse IgG-specific secondary antibody and ECL.
DOI: https://doi.org/10.7554/eLife.43322.009

established pattern of Kv2.1 immunolabeling in these neurons (*e.g.*, *Bishop et al., 2015*; *Trimmer, 1991*; *Rhodes et al., 1995*) and that overlaps precisely with simultaneous immunolabeling obtained with the native K89/34 mAb (*Figure 6C*). The D3/71R R-mAb also yields an immunoreactive band on immunoblots similar to that obtained with K89/34 and K89/34R (*Figure 6D*). These results suggest that the mAb cloning approach described here can be used to effectively recover functional R-mAbs from non-viable hybridomas and could be applied by researchers that have such hybridomas in their collections.

## Discussion

In this study, we developed a novel R-mAb generation and validation procedure that we used to generate a valuable R-mAb resource by converting a library of widely used mouse mAbs into recombinant reagents. Our approach was adapted from that of Gavin Wright and colleagues (*Crosnier et al., 2010*) with modifications to facilitate higher throughput cloning and validation steps. The efficiency of the process was further enhanced by restriction digest of $V_L$ PCR products to eliminate the MOPC-21 derived aberrant light chain. R-mAbs were produced in the culture medium from transiently transfected cells under standard mammalian cell culture conditions in sufficient quantities that they did not require purification for effective use in IB, ICC, and IHC assays. We were also able to recover functional R-mAbs from a cryopreserved hybridoma cell line that was no longer viable in cell culture *via* RNA extraction, cloning and expression. Moreover, in the process we generated subclass-switched R-mAbs that are more amenable to multiplex labeling in IB, ICC, and IHC applications. We then employed these mAbs in multiplex IHC experiments in combinations that were not possible prior to their cloning and expression as subclass-switched R-mAbs. Using the method described here, most researchers with standard molecular biology expertise can generate functional R-mAb expression plasmids with the advantages associated with recombinant reagents from their own cryopreserved hybridomas, whether viable or non-viable. Moreover, producing R-mAbs from cells transfected with any of the large collection of expression plasmids whose generation is described here is within the reach of anyone with standard mammalian cell culture capabilities.

We previously undertook a systematic analysis of anti-mouse IgG subclass-specific secondary antibodies and demonstrated that the use of mouse mAbs of different IgG subclasses in combination with anti-mouse IgG subclass-specific secondary antibodies for simultaneous multiplex immunolabeling allows for robust and specific labeling in several immunolabeling applications, and also provides enhanced signal-to-noise (background) in immunolabeling of brain samples when compared to generic anti-mouse IgG (H + L) secondary Abs (*Bekele-Arcuri et al., 1996*; *Rhodes et al., 1997*; *Lim et al., 2000*; *Rasband et al., 2001*; *Manning et al., 2012*). Simultaneous multiplex labeling of multiple targets within a single sample allows for better comparison of their relative levels, co-localization, and overall tissue architecture. Multiplex labeling also conserves valuable samples, as multiple targets can be interrogated in the same sample instead of across multiple samples, which can be a particular concern in studies using human tissue samples that may be available in limited quantity. However, multiplex labeling using the classical approach of employing primary Abs raised in different species and their subsequent detection with species-specific secondary Abs is highly limited by the lack of available Abs from numerous species. As one example, interrogating the Abs registered with the Antibody Registry (2,381,068 as of 9/9/18) shows a preponderance (84% in total) of Abs raised in only three species: rabbits (44.6%), mice (31.2%) and goats (8.2%), with all other species together accounting for the remaining 16%. Among the predominant species, goat Abs are exclusively polyclonal and as such cannot be used with one another in simultaneous multiplex labeling. Moreover, the bulk of secondary Abs are raised in goats, constraining facile detection of primary Abs in multiplex labeling experiments employing goat primary Abs. Rabbit Abs (whether poly- or mono-clonal) have limited utility for multiplex labeling, as unlike most other mammals, rabbits do not make IgGs of distinct subclasses but only a single generic IgG (*Knight et al., 1985*). As such employing multiple goat or rabbit primary Abs for multiplex immunolabeling involves costly and time-consuming procedures such as direct labeling with different fluorophores or sequential multiplexing technologies such as Opal (*Stack et al., 2014*).

While the utility of mouse mAbs (and less common rat mAbs) for simultaneous multiple immunolabeling is enhanced by the availability of subclass-specific secondary antibodies, it remains that

mouse mAbs against any given target may be available in only a single IgG subclass. In general, mouse mAb collections reflect the representation of these subclasses in the circulating serum IgGs in immunized BALB/c mice, which is ≈70% IgG1, ≈20% IgG2a and ≈10% IgG2b (*Natsuume-Sakai et al., 1977*). As a prominent example, the large mouse mAb catalog of ThermoFisher Scientific (10,992 independent IgG mouse mAbs as of 6/28/18) follows this remarkably closely, comprising 69% IgG1, 18% IgG2a and 12% IgG2b (Matt Baker, ThermoFisher Scientific, personal communication). As such, the flexibility of using mouse mAbs in multiplex labeling is restricted by the prevalence of those of the IgG1 subclass. It is possible to selectively screen for mAbs of the rarer IgG2a and IgG2b subclasses of mouse IgGs in the process of their development and screening (*Gong et al., 2016*; *Liu et al., 2015*). It is also possible, although labor-intensive, to manipulate hybridomas in culture and select and/or screen for subclass switched hybridoma-generated mAbs (*Faguet and Agee, 1993*). However, it remains that the optimal utility of mouse mAbs in research and diagnostics is limited by the preponderance of IgG1 mAbs. Here, we have enhanced the flexibility of a substantial fraction of an existing library of widely used mouse mAbs by converting them to subclass-switched R-mAbs without altering target binding specificity. We show using mixtures of mAbs and R-mAb that we can obtain effective simultaneous multiplex labeling for combinations that were previously unattainable due to conflicting IgG subclass.

Antibody reformatting in our system is easily accomplished by the choice of plasmid backbone used for the original cloning, or by subcloning the $V_L$-joining fragment-$V_H$ cassette (*Figure 1B*) between plasmids with distinct $C_H$ domains. Ultimately, one can envision using this approach to convert any mouse mAb into the corresponding set of mouse IgG1, IgG2a, and IgG2b subclass R-mAb expression plasmids. Multiplex labeling could be further expanded by constructing IgG expression vectors containing $C_H$ regions of other species that have IgG subclasses such as rat and human (but not rabbit, in which all IgG Abs are of a generic IgG class), for which subclass-specific secondary antibodies are also widely available. For example, we interrogated the ThermoFisher secondary Ab catalog, which contains 52 different IgG subclass-specific secondary antibodies for mouse IgG subclasses, 47 for rat IgG subclasses, and 43 for human IgG subclasses. Together, the wide availability of such a broad range of reliable secondary antibodies with distinct conjugates provides ample flexibility for specific detection of multiple R-mAbs with distinct $C_H$ regions engineered for separate detection in multiplex labeling experiments.

Previous attempts by others to generate R-mAbs have been complicated by the MOPC-21 derived aberrant kappa light chain that is present in many widely used myeloma fusion partners (*Crosnier et al., 2010*; *Cochet et al., 1999*; *Duan and Pomerantz, 1994*; *Yuan et al., 2004*). For example, Crosnier et al., (*Crosnier et al., 2010*) showed that depending on the hybridoma, 26% to 70% of R-mAb plasmids they generated contained this aberrant sequence. We have incorporated a simple restriction enzyme digest into our process to eliminate the aberrant kappa light chain (*Juste et al., 2006*). For most of our hybridomas, the majority of the $V_L$ PCR product was cleaved by BciVI, indicating that the MOPC-21 aberrant light chain comprised the greatest fraction of total light chain amplicons. By eliminating this transcript, the overall screening and cloning burden for each mAb was reduced substantially. However, depending on the particular hybridoma, the efficiency of generating clones that produced functional R-mAbs even after BciVI digestion of the $V_L$ domain PCR products was still highly variable. This was evidenced by the overall efficiency of slightly below 50% for the successful projects tabulated in *Supplementary file 1*, and also by the projects that were unsuccessful on the first pass in which clones passed the colony PCR and restriction digest screening steps but failed in the COS-ICC screening assay for functional R-mAbs. Moreover, a lack of allelic exclusion at heavy and light chain IgG loci has been documented in hybridomas, resulting in expression of multiple IgG heavy and/or light chains at the mRNA and protein levels (*Zack et al., 1995*; *Blatt et al., 1998*; *Bradbury et al., 2018*; *Ruberti et al., 1994*). In these previous studies, while multiple combinations of heavy and light IgG chains were produced by monoclonal hybridoma cells, only one combination of heavy and light IgG chains present in the hybridoma produced an R-mAb capable of recognizing the target antigen. Varying degrees of allelic exclusion in the hybridomas that served as the source of our R-mAbs could also contribute to the variable efficiency of whether colony PCR and restriction enzyme assay-positive clones would produce an R-mAb that recapitulated immunolabeling obtained with the native mAb. Technical issues, such as PCR-induced mutations may have also contributed to the generation of R-mAb clones that failed in the COS-IF-ICC assay. We did not systematically analyze these negative clones, so we do not know whether these plasmids

contain residual aberrant light chains from the BciVI digestion step, plasmids containing other heavy and/or light chains, or mutated forms of *bona fide* R-mAb chains with impaired/lost functionality. That the vast majority (84%) of the 114 projects for which $\geq 4$ clones were evaluated yielded at least one positive R-mAb suggests that the presence of a small and variable number of negative clones does not impact the overall success of the approach described here, given the simple but effective nature of microplate screening assays, such as the ICC assay employed here, to rapidly and easily screen candidate R-mAbs.

There is a growing impetus to enhance research reproducibility by both improving the quality of Abs used in basic research and in diagnostics, and the transparency of their reporting as related to the exact molecularly-defined Ab that was used (*Taussig et al., 2018*; *Bradbury and Plückthun, 2015*; *Uhlen et al., 2016*). Using mAbs in their recombinant form and defining them unambiguously at the molecular level by publishing their $V_L$ and $V_H$ domain sequences is a step towards achieving this goal and elevate reproducibility world-wide. Among the different forms of research and diagnostic Abs, polyclonal Ab preparations have unique benefits (*Ascoli and Aggeler, 2018*), due to the presence of Abs recognizing distinct epitopes on the target protein. However, in any of the diverse forms in which they are available (antiserum, IgG fractions, affinity-purified) polyclonal Abs are inherently limited by their intrinsic lack of precise molecular definition. The inability to rigorously define the clonal composition of specific preparations can contribute to batch-to-batch variation in the efficacy and specificity of the polyclonal collection for any specific application, negatively impacting the reproducibility of research performed with such Abs. Moreover, unlike mAbs, polyclonal Abs are a finite resource whose depletion can lead to batch-to-batch variability or complete lack of availability, stymieing independent reproduction of research using the same reagent. Conventional mAbs are advantageous in lacking the molecular complexity of polyclonal preparations, and in being renewable reagents. However, the conversion of conventional hybridoma-generated mAbs into R-mAbs offers numerous additional advantages. This includes ensuring their broad availability by generating a publicly accessible DNA sequence archive that can be used to synthesize the functional $V_L$ and $V_H$ domain sequences. R-mAbs are also more easily distributed as plasmids or DNA sequence, a feature that should enhance dissemination of this resource, relative to the more labile cryopreserved and/or living hybridomas that are less amenable to long-distance transport. Cloning also enhances research reproducibility by the unambiguous definition of mAbs at the level of their cDNA sequence, as well as the enhanced control of molecular composition afforded by expressing a single light and heavy chain combination from transfected cells and their high-level animal-free production in cell culture. Cloning of mAbs also allows for increased utility as offered by their subsequent engineering into alternate forms, such as the subclass-switched R-mAbs generated here. The method described here represents a straightforward approach to producing functional R-mAbs from existing murine hybridomas that could be applied to other valuable mAb collections to ensure their permanent archiving, which cannot be assured when they exist solely in the form of hybridomas cryopreserved in liquid nitrogen. In particular, there are many cases of important hybridomas, including entire collections, that have been discarded upon the closure of the laboratory that formerly housed and financially supported the maintenance of the cryopreserved collection. The enhanced utility, permanence, and cost-effectiveness of R-mAbs as generated by the approach described here is one route to circumvent the negative impact to research effectiveness and reproducibility that such losses represent. Simple conversion of mAbs to R-mAbs, as described here, also paves the way for higher throughput approaches, for example those employing high throughput 'next generation' sequencing to simultaneously obtain light and heavy chain sequences from large pools of hybridomas (*Chen et al., 2018*). In theory this should allow for larger scale '*en masse*' (as opposed to the mAb-by-mAb approach used here) conversion efforts more suitable to large hybridoma collections. For example, our larger collection of $\approx 40,000$ target-specific mAbs archived in our cryopreserved hybridoma collection, and other widely used collections, such as that housed at the Developmental Studies Hybridoma Bank (http://dshb.biology.uiowa.edu/; comprising 3,672 mAbs as of 9/9/18) might represent appropriate targets for such next generation sequencing approaches. Future efforts to generate R-mAbs from cryopreserved hybridoma collections will further enhance the rigor, reproducibility and overall effectiveness of Ab-based research.

# Materials and methods

**Key resources table**

| Reagent type (species) or resource | Designation | Source or reference | Identifiers | Additional information |
|---|---|---|---|---|
| Cell line (*Cercopithecus aethiops*) | COS-1 | ATCC Cat No CRL-1650; PMID: 6260373 | RRID:CVCL_0223 | |
| Antibody | numerous | | | See Table 2, *Supplementary file 1*, *2* |
| Recombinant DNA reagent | p1316 plasmid | PMID: 20525357 | | |
| Software | Photoshop | Adobe Systems | RRID:SCR_014199 | |
| Software | Axiovision | Carl Zeiss MicroImaging | RRID:SCR_002677 | |
| Software | Fiji | PMID: 22743772 | RRID:SCR_002285 | |

## Primers

Primer sets for mouse Ig V region amplification and fusion PCR (F-PCR) were used as described previously (*Crosnier et al., 2010*; *Müller-Sienerth et al., 2014*). The following primers were used for other PCR steps.

Amplification of the Joining Fragment (*Crosnier et al., 2010*)

Primer 21: 5'- GGGCTGATGCTGCACCAACTGTA-3'

Primer 26: 5'-ACTGCTTGAGGCTGGACTCGTGAACAATAGCAGC-3'

Colony PCR:

UpNotI: 5'-TTTCAGACCCAGGTACTCAT-3'

DownAscI: 5'-GGGCAGCAGATCCAGGGGCC-3' (reverse primer for IgG1 vector)

Rev IgG2a: 5'- ACCCTTGACCAGGCATCCTAGAGT- 3' (reverse primer for IgG2a vector)

Mouse $\gamma$2a $C_H$ domain amplification (restriction sites are underlined):

IgG2a-F-AscI: 5'-ATATCAC<u>GGCGCGCCC</u>AACAGCCCCATCGGTCTATCCA-3'

IgG2a-R-XbaI: 5'GACTGA<u>TCTAGA</u>TCATTTACCCGGAGTCCGGGAGAA-3'

R-mAb Sequencing:

Forward strand of $V_L$ region: UpNotI = 5'-TTTCAGACCCAGGTACTCAT-3'

Reverse strand of $V_L$ region (IgG2a plasmids): Seq_VL_Rev_IgG2a = 5' - CCAACTGTTCAG-GACGCCATT −3'

Forward strand of VH region: VH_seq_Forward = 5'- TCCCAGGCCACCATGAA −3'

Reverse strand of $V_H$ region (IgG1 plasmids): DownAscI = 5'-GGGCAGCAGATCCAGGGGCC-3'

Reverse strand of $V_H$ region (IgG2a plasmids): Rev IgG2a = 5'- ACCCTTGACCAGGCATCCTAGAGT- 3'

## RNA preparation from cryopreserved hybridomas and RT-PCR

The Ambion RNAqueous 96 Extraction kit (Thermo Fisher Cat# AM1920) was used for high through-put RNA extraction. Frozen vials containing $0.5–1 \times 10^7$ hybridoma cells per vial were thawed in a 37°C water bath for 5 min, in batches of 20 vials for high through-put purposes. Cells were spun down in a table top centrifuge at 2000 rpm for 5 min, the supernatant was removed, and cells were washed with 1 mL cold PBS. A 250 µL aliquot of this cell suspension, representing $1–3 \times 10^6$ cells, was used for RNA extraction according to the manufacturer's instructions. We replica plated using 4.0 µL of RNA (1/16 of the extract volume) to a second 96 well plate and used the Superscript III Reverse Transcriptase First Strand Synthesis System (Thermo Fisher Cat# 18080051) for high throughput cDNA synthesis using oligo (dT) primers.

## Immunoglobulin V region amplification, BciVI treatment, and fusion PCR

Following the cDNA synthesis reaction, we replica plated ≈5% of the volume (1.0 µL of 10-fold dilution) of the cDNA synthesis product to a third 96 well plate to serve as the template for PCR amplification of the IgG $V_H$ and $V_L$ domain sequences. As such, the templates for the $V_H$ and $V_L$ domain

PCR amplification represented the cDNA yield from ≈3,000–7,500 hybridoma cells. Amplification of Ig V region sequences and fusion PCR (F-PCR) to join $V_L$ and $V_H$ PCR products were performed as described (*Crosnier et al., 2010*; *Müller-Sienerth et al., 2014*) with the noted modifications. Briefly, degenerate primer sets were used to amplify mouse Ig kappa $V_L$ and $V_H$ sequences using PFU Ultra II Fusion HS DNA polymerase (Agilent Technologies Cat# 600670) and Advantage 2 Polymerase Mix (ClonTech Cat# 639201) for $V_L$ and $V_H$ amplification, respectively. PCR conditions for $V_L$ amplification were: 95°C for 5 min; 5 cycles of 95°C for 20 s, 60°C for 20 s, 72°C for 30 s; 19 cycles of 95°C for 20 s, 60.5°C for 20 s with 0.5°C decrement per cycle, 72°C for 30 s; 10 cycles of 95°C for 20 s, 55°C for 20 s, 72°C for 30 s; 72°C for 15 min. PCR conditions for $V_H$ amplification were: 95°C for 5 min; 5 cycles of 95°C for 45 s, 62°C for 30 s, 72°C for 1 min; 19 cycles of 95°C for 45 s, 64.5°C for 30 s with 0.5°C decrement per cycle, 72°C for 1 min; 10 cycles of 95°C for 45 s, 55°C for 30 s, 72°C for 1 min; 72°C for 15 min. $V_L$ PCR products (7.0 μL per reaction) were digested with 5 units of the restriction enzyme BciVI (BfuI) (Thermo Fisher Cat# ER1501) in a 20 μL reaction at 37°C for 2 hr. The enzyme was inactivated by heating to 80°C for 20 min.

In preparation for fusion of the $V_L$ and $V_H$ PCR products, a joining fragment was produced using the mouse Ig expression plasmid P1316 (a gift of Dr. Gavin Wright, Sanger Institute, Cambridge, UK, now available from Addgene as plasmid #28217). The joining fragment contains the following sequences (5' to 3'): mouse kappa light chain constant region, a polyadenylation signal, a CMV promoter, and the leader sequence for the Ig heavy chain. P1316 was used as a template in a 50 μL PCR containing 0.2 μM of primers 21 and 26, 0.2 mM dNTPs, and 1.0 μL PFU Ultra II Fusion HS DNA polymerase. PCR conditions were: 95°C for 5 min; 5 cycles of 95°C for 20 s, 60°C for 20 s, 72°C for 45 s; 19 cycles of 95°C for 20 s, 60.5°C for 20 s with 0.5°C decrement per cycle, 72°C for 45 s; 10 cycles of 95°C for 20 s, 55°C for 20 s, 72°C for 45 s; 72°C for 15 min. The 1.7 kb joining fragment was purified (Qiagen/QiaQuick PCR Purification Cat# 28106) in preparation for F-PCR.

$V_L$ (BciVI restriction enzyme digested), the joining fragment, and $V_H$ PCR products were joined via F-PCR in a 96-well format. We observed that purification of $V_L$ and $V_H$ PCR products was not necessary. Each 50 μL reaction consisted of the following: 0.2 μM of primers 51 and 52 (*Crosnier et al., 2010*; *Müller-Sienerth et al., 2014*), 0.2 mM dNTPs, 1.5 μL $V_L$ (BciVI digested), 0.5 μL $V_H$, 0.5 μL purified joining fragment (50 ng), and 1.0 μL PFU Ultra II Fusion HS DNA polymerase. PCR conditions were: 95°C for 2 min; 11 cycles of 95°C for 45 s, 63°C for 30 s, 72°C for 5 min; 7 cycles of 95°C for 45 s, 62°C for 30 s with 1°C decrement per cycle, 72°C for 5 min, 95°C for 45 s; 26 cycles of 56°C for 30 s, 72°C for 5 min; 72°C for 15 min.

## Cloning of immunoglobulin variable domain regions into a dual promoter expression plasmid

F-PCR products were digested with FastDigest NotI and AscI (Thermo Fisher Cat# ER0595 and ER1891, respectively) at 37°C for 20 min, followed by inactivation at 80°C for 5 min and column purification (Qiagen/QiaQuick PCR Purification Cat# 28106). The P1316 plasmid was also NotI/AscI digested and gel purified (Qiagen/QiaQuick Gel Extraction Cat# 28706). P1316 is a derivative of the pTT3 expression vector (*Durocher et al., 2002*) and consists of (5' to 3'): a CMV promoter, the mouse V kappa leader sequence, a NotI restriction site, an insert consisting of $V_L$/joining fragment/$V_H$, and the mouse IgG1 $C_H$ sequence amplified from mouse genomic DNA, flanked by AscI and XbaI restriction sites (*Crosnier et al., 2010*; *Müller-Sienerth et al., 2014*). Ligation was performed overnight at 16°C with T4 DNA ligase (Thermo Fisher Cat# 15224017) using 20 ng insert and 20 ng vector, a 3:1 molar ratio. Half of each ligation reaction was used to transform 25 μL of Mach I chemically competent *E. coli.* (Thermo Fisher Cat# C862003). Cells and DNA were incubated on ice for 30 min, heat shocked at 42°C for 30 s, incubated on ice for 2 min, and allowed to recover for 1.0 hr in 250 μL SOC medium in a 37°C shaking incubator. The cells were spun in a centrifuge at 3000 rpm (≈950 x g) for 2 min and the supernatant was removed until 150 μL remained. Cells were resuspended, and the entire volume was plated on LB plates containing 100 μg/mL ampicillin and incubated overnight at 37°C.

## Colony PCR, restriction analysis and sequencing of R-mAb clones

Colony PCR was performed to identify *E. coli* colonies that contained the full-length, 2.4 kb Ig cassette. Colonies were diluted in 96-well plates containing 50 μL water and patch plates were made

for later recovery of positive clones. 2 µL of diluted colony was used in each PCR. Conditions were 94°C for 5 min; 23 cycles of 94°C for 20 s, 58°C for 30 s, 72°C for 2.5 min; 72°C for 10 min. For additional confirmation of the presence of the full-length Ig cassette, plasmid DNA was isolated from PCR positive clones and subjected to restriction enzyme digestion with NotI/AscI at 37°C for 20 min followed by agarose gel electrophoresis. The $V_L$ and $V_H$ regions of functional R-mAbs were subjected to sequencing in both orientations to generate a permanent archive. The primers 'UpNotI' and 'Seq_VL_Rev_IgG2a' were used for $V_L$ domain sequencing, and the 'VH seq forward', and either 'DownAscI' or 'Rev IgG2a' for sequencing of $V_H$ regions in the IgG1 or IgG2a expression plasmids, respectively.

## Generation of a mouse IgG2a expression vector

The mouse γ2a $C_H$ domain was amplified from the cDNA preparation that was obtained from the K28/43 (RRID: AB_2292909) hybridoma (*Rasband et al., 2002*; *Shibata et al., 2003*) that was used for cloning of the K28/43 $V_L$ and $V_H$ domains. PCR conditions were: 94°C for 5 min; 29 cycles of 94°C for 30 s, 65°C for 30 s, 72°C for 30 s. The forward primer included an AscI restriction site and the reverse primer included an XbaI restriction site to facilitate cloning into the K28/43 IgG1 recombinant R-mAb plasmid. The K28/43 IgG1 recombinant R-mAb plasmid was derived from the P1316 plasmid (*Crosnier et al., 2010*) by restriction enzyme-based cloning of K28/43 variable region sequences as described above. The IgG2a $C_H$ PCR product and the K28/43 IgG1 recombinant plasmid were both digested with AscI and XbaI restriction enzymes (New England BioLabs Cat# R0558 and R0145, respectively) and column purified (Qiagen/QiaQuick PCR Purification Cat# 28106) or agarose gel purified, respectively (Qiagen/QiaQuick Gel Extraction Cat# 28706). Because XbaI is methylation sensitive, the K28/43 IgG1 plasmid was sourced from $dam^-/dcm^-$ *E. coli* (New England Biolabs Cat# C2925). T4 DNA ligase (New England Biolabs Cat# M0202) was used to insert the IgG2a $C_H$ fragment into the digested K28/43R plasmid to generate a K28/43 IgG2a R-mAb plasmid, which was confirmed by DNA sequencing.

## R-mAb expression in mammalian cells

COS-1 cells (ATCC Cat No CRL-1650; RRID:CVCL_0223) were used for R-mAb expression. To rule out inter-species contamination, cells were authenticated at ATCC as being from African Green Monkey by a PCR based method to detect species-specific variants of the cytochrome C oxidase I gene (COI analysis). Cells were tested in house for mycoplasma contamination using the MycoAlert Mycoplasma Detection Kit (Lonza Catalog#: LT07-318). For production of recombinant mAbs in mammalian cell culture, $3 \times 10^5$ COS-1 cells were plated on 35 mm tissue culture dishes and cultured overnight in DMEM (high glucose/pyruvate, Thermo Fisher Cat# 11995065) with 10% Fetal Clone III (HyClone Cat# SH30109.03) and 100 µg/ml penicillin/streptomycin (Thermo Fisher Cat# 15140122). Cells were then transfected with a 1:1 ratio of plasmid (1 µg):Lipofectamine 2000 (1 µL) (Thermo Fisher Cat# 11668019) diluted in Opti-MEM reduced serum medium (Thermo Fisher Cat# 31985070). Following an overnight incubation, the transfection solution was replaced with culture medium, and the cells incubated for an additional 3–6 days, at which time the conditioned medium was collected as R-mAb tissue culture supernatant (TC supe) for analysis. In certain cases, R-mAbs were produced from COS-1 cells cultured in 12 well plates, using proportionally reduced amounts of cells ($1.5 \times 10^5$) plasmid (0.5 µg):Lipofectamine 2000 (0.5 µL) and Opti-MEM.

## COS-1 cell immunofluorescence immunocytochemistry validation assay

R-mAb TC supes were screened for immunoreactivity in an immunofluorescence assay against transiently transfected COS-1 cells cultured in 96-well plates. COS-1 cells were plated in black, clear bottom 96-well plates (Greiner Cat# 655090) at a density of 4,700 cells/well. After overnight incubation, each well received 50 ng plasmid DNA encoding the R-mAb target protein plus Lipofectamine 2000 at a 1:1 ratio as described above. On day three post-transfection, cells were washed three times with DPBS (138 mM NaCl, 2.67 mM KCl, 1.47 mM $KH_2PO_4$, 8.1 mM $Na_2HPO_4$, 1 mM $CaCl_2$ and 1 mM $MgCl_2$), pH 7.4 and then fixed using 3.0% formaldehyde (prepared fresh from paraformaldehyde) in in DPBS plus 0.1% Triton X-100 on ice for 20 min. Cells were washed three times with DPBS/0.1% Triton X-100, blocked with Blotto/0.1% Triton X-100 for 1 hr, and stored in DPBS/0.02% sodium azide. For primary antibody labeling, R-mAb TC supes were used without dilution and

hybridoma-generated mAb TC supe controls (see *Table 2* for details of non-R-mAb Abs used in this study) were diluted 1:10 in COS-1 cell culture medium. Each R-mAb was tested alone and in combination with the corresponding hybridoma-generated mAb TC supe. Primary antibodies were incubated at room temperature for 1 hr and cells were washed 3 × 10 min with Blotto/0.1% Triton X-100. Secondary labeling was performed at room temperature for 30 min using subclass-specific, anti-mouse secondary antibodies conjugated to Alexa Fluors (Thermo Fisher, Cat#/IgG subclass/ Alexa Fluor dye conjugates: (A-21121/IgG1/488 and A-21241/IgG2a/647) and diluted to 1.3 µg/mL in Blotto/0.1% Triton X-100. Hoechst 33342 (Thermo Fisher Cat# H3570) was used at 0.1 µg/mL in the secondary antibody cocktail to stain nuclear DNA. Cells were washed 3 × 10 min with DPBS/ 0.1% Triton X-100. Imaging was performed using a Zeiss M2 AxioImager microscope. Images were processed using Axiovision (Carl Zeiss Microimaging, RRID:SCR_002677 and Fiji (NIH, RRID:SCR_ 002285) software.

For higher resolution imaging, COS-1 cells were plated on poly-L-lysine coated #1.5 glass cover slips and cultured overnight followed by transfection with plasmids encoding the target protein. Cells were fixed and immunolabeled as described in the previous section. Images were acquired on a Zeiss AxioImager M2 microscope using a 40x/0.8 NA plan-Apochromat oil-immersion objective and an AxioCam MRm digital camera. Optical sections were acquired using an ApoTome two

**Table 2.** Non-R-mAb antibodies used in this study.

Table lists Abs used in this study outside of the R-mAbs whose generation is described here. For each Ab the name, immunogen used in Ab generation, source and RRID number in the Antibody Registry, form and concentration/dilution used, and specific use in this paper is detailed.

| Antibody | Immunogen | Manufacturer information | Concentration/dilution used | Figures |
|---|---|---|---|---|
| KC | Synthetic peptide aa 837–853 of rat Kv2.1 | Rabbit pAb, In-house (Trimmer Laboratory), RRID: AB_2315767 | Affinity purified, 1:100 | 4 |
| PSD-95 | Fusion protein aa 77–299 of human PSD-95 | Rabbit pAb, In-house (Trimmer Laboratory), RRID: AB_2750832 | Affinity purified, 1:150 | 3 |
| K28/43 | Fusion protein aa 77–299 of human PSD-95 | Mouse IgG2a mAb, NeuroMab RRID:AB_10698024 | Tissue culture supernatant, 1:5 | 3 |
| K28/86 | Fusion protein aa 77–299 of human PSD-95 | Mouse IgG1 mAb, NeuroMab RRID:AB_10698179 | Tissue culture supernatant, 1:5 | 3 |
| K57/1 | Synthetic peptide aa 209–225 of human Kv4.2 | Mouse IgG1 mAb, NeuroMab RRID:AB_10672254 | Tissue culture supernatant, 1:5 | 4 |
| K65/35 | Fusion protein aa 1308–1381 of rat CASPR | Mouse IgG1 mAb, NeuroMab, RRID:AB_10671175 | Tissue culture supernatant, 1:5 | 4 |
| K89/34 | Synthetic peptide aa 837–853 of rat Kv2.1 | Mouse IgG1 mAb, NeuroMab RRID:AB_10672253 | Tissue culture supernatant, 1:10 | 6 |
| N28/9 | Fusion protein aa 492–560 of rat VGluT1 | Mouse IgG1 mAb, NeuroMab RRID:AB_10673111 | Tissue culture supernatant, 1:5 | 4 |
| N87/25 | Fusion protein aa 370–433 of mouse GABA-A-receptor β3 subunit | Mouse IgG1 mAb, NeuroMab RRID:AB_10673389 | Tissue culture supernatant, 1:5 | 4 |
| N106/65 | Full-length recombinant human Ankyrin-G | Mouse IgG2a mAb, NeuroMab RRID:AB_10673449 | Tissue culture supernatant, 1:5 | 4 |
| N147/6 | Fusion protein aa 1–341 (full-length) of human QKI-5 | Mouse IgG2b mAb, NeuroMab RRID:AB_10671658 | Tissue culture supernatant, 1:5 | 4 |
| N206A/8 | Synthetic peptide aa 411–422 of human GFAP | Mouse IgG1 mAb, NeuroMab RRID:AB_10672298 | Tissue culture supernatant, 1:5 | 4 |

DOI: https://doi.org/10.7554/eLife.43322.010

structured illumination system (Carl Zeiss MicroImaging). Imaging and post processing were performed in Axiovision and Photoshop (Adobe Systems; RRID:SCR_014199).

## Multiplex immunofluorescence labeling of immunoblots

Multiplex immunofluorescence labeling of immunoblots using mouse IgG subclass-specific secondary antibodies was performed as described previously (*Manning et al., 2012*). In brief, samples were generated from COS-1 cells transiently transfected to express individual target proteins essentially as described above for the immunofluorescence experiments except that the cells were cultured in 35 mm tissue cultures dishes. Transfected COS-1 cells were washed once with ice-cold PBS and lysed with 150 µL of ice-cold lysis buffer containing 1% v/v Triton X-100, 150 mM NaCl, 1 mM EDTA, 50 mM Tris-HCl (pH 7.4), 1 mM sodium orthovanadate, 5 mM NaF, 1 mM PMSF and a protease inhibitor cocktail for 10 min at 4°C (*Shi et al., 1994*). The cell lysates were centrifuged at 12,000 x g at 4 °C for 10 min. The cell lysate supernatants were mixed with 150 µL of 2X RSB and size-fractionated by 7.5% SDS–PAGE. Following SDS-PAGE, proteins were transferred to nitrocellulose membranes (Bio-Rad Cat# 1620115), which were blocked for 1 hr with Blotto (3% w/v nonfat milk in Tris-buffered saline (TBS: 50 mM Tris, pH 7.5, 150 mM NaCl) plus 0.1% v/v Tween-20 followed by 2 hr or overnight incubation with primary antibodies. Primary antibodies were mAb TC supes diluted 1:10, non-diluted R-mAb TC supes, and an in-house anti-PSD-95 rabbit polyclonal antibody raised against a GST fusion protein, GSTKAP1.13, containing amino acids 77–299 of human PSD-95 [clone 2, (*Kim et al., 1995*)] and that crossreacts with SAP97 (see *Table 2* for details of non-R-mAb Abs used in this study). After three washes with Blotto, the membranes were incubated with the appropriate subclass-specific Alexa Fluor conjugated secondary antibodies (*Manning et al., 2012*) for 1 hr. After three washes with TBS containing 0.1% v/v Tween-20, the immunoblots were visualized directly on a FluorChem Q imager (Cell Biosciences Cat# DE500-FCQ). Alternatively, crude rat brain membranes (RBM) (*Shi et al., 1994*) were subjected to immunoblotting as described above except that a single RBM sample (3 mg protein) were size fractionated on a curtain gel, and after transfer to nitrocellulose the membrane was cut into 30 strips, each containing 100 µg of RBM protein. Immunolabeling was detected on autoradiography film after treatment of strip immunoblots with HRP-conjugated anti-mouse IgG-specific secondary antibody and enhanced chemiluminescence (ECL).

## Multiplex immunofluorescence labeling of brain sections

Multiplex immunofluorescence labeling of rat brain sections was performed essentially as described previously (*Manning et al., 2012*; *Bishop et al., 2015*). All experimental procedures were approved by the UC Davis Institutional Animal Care and Use Committee and conform to guidelines established by the National Institutes of Health (NIH). Rats were anesthetized with sodium pentobarbital (Fatal-Plus solution, 100 mg/kg sodium pentobarbital) and perfused transcardially with 100 mL of phosphate buffered saline (PBS), containing 10 units/mL heparin, pH 7.4, followed by 400 mL of 4% formaldehyde (prepared fresh from paraformaldehyde) in 0.1 M sodium phosphate buffer or PB (pH 7.4). The brains were removed, cryoprotected for 48 hr in 30% sucrose, frozen in a bed of pulverized dry ice, and then cut into 30 µm sections on a freezing-stage sliding microtome. Sections were collected in 0.1 M PB and processed immediately for immunohistochemistry. Free-floating brain sections were blocked with 10% goat serum in 0.1 M PB containing 0.3% Triton X-100 (vehicle) for 1 hr at RT and then incubated overnight at 4°C in vehicle containing different combinations of primary antibodies (see *Table 2* for details of non-R-mAb Abs used in this study). The following day sections were washed 4 × 5 min each with vehicle, and then incubated for 1 hr at RT in mouse IgG subclass-specific or anti-rabbit Alexa-conjugated secondary antibodies as described previously (*Manning et al., 2012*; *Strassle et al., 2005*). Sections were then washed 2 × 5 min each with 0.1 M PB, 2 × 5 min each with 0.05 M PB and mounted on gelatin-coated microscope slides and air dried. Sections were cover slipped after adding ProLong Gold Antifade Mountant (Thermo Fisher Scientific catalog # P36930). Images were obtained on a Zeiss Axiovert 200 microscope with Apotome. Imaging and post-imaging processing was performed in Zeiss Axiovision and Adobe Photoshop software, taking care to maintain any linear differences in signal intensities present in the original samples.

## Acknowledgments

This work was funded by NIH research grants U24 NS050606, R24 NS092991 and U24 NS109113 to JS Trimmer. We thank the current and former staff members of the UC Davis/NIH NeuroMab Facility and the Trimmer laboratory for their contributions and dedicated efforts, and Dr. Randall Stewart at the National Institute of Neurological Disorders and Stroke for support and helpful advice. We thank Dr. Gavin Wright of the Sanger Institute for his generous gift of the P1316 expression plasmid and helpful advice.

## Additional information

### Funding

| Funder | Grant reference number | Author |
| --- | --- | --- |
| National Institute of Neurological Disorders and Stroke | U24 NS050606 | James S Trimmer |
| National Institute of Neurological Disorders and Stroke | R24 NS092991 | James S Trimmer |
| National Institute of Neurological Disorders and Stroke | U24 NS109113 | James S Trimmer |

The funders had no role in study design, data collection and interpretation, or the decision to submit the work for publication.

### Author contributions

Nicolas P Andrews, Conceptualization, Resources, Data curation, Formal analysis, Supervision, Funding acquisition, Validation, Investigation, Methodology, Writing—original draft, Project administration, Writing—review and editing; Justin X Boeckman, Conceptualization, Data curation, Formal analysis, Validation, Investigation, Methodology, Writing—original draft, Writing—review and editing; Colleen F Manning, Joe T Nguyen, Hannah Bechtold, Conceptualization, Data curation, Formal analysis, Validation, Investigation, Methodology, Writing—review and editing; Camelia Dumitras, Conceptualization, Data curation, Validation, Investigation, Methodology; Belvin Gong, Supervision, Validation, Investigation, Methodology; Kimberly Nguyen, Data curation, Supervision, Validation, Investigation, Methodology; Deborah van der List, Data curation, Formal analysis, Investigation, Methodology; Karl D Murray, Conceptualization, Data curation, Formal analysis, Supervision, Validation, Investigation, Methodology; JoAnne Engebrecht, James S Trimmer, Conceptualization, Resources, Data curation, Formal analysis, Supervision, Funding acquisition, Validation, Investigation, Methodology, Project administration, Writing—review and editing

### Author ORCIDs

Justin X Boeckman  http://orcid.org/0000-0002-0022-1474
Joe T Nguyen  http://orcid.org/0000-0002-6647-0561
James S Trimmer  http://orcid.org/0000-0002-6117-3912

### Ethics

Animal experimentation: This study was performed in strict accordance with the recommendations in the Guide for the Care and Use of Laboratory Animals of the National Institutes of Health. All of the animals were handled according to approved institutional animal care and use committee (IACUC) protocols (#20485) of the University of California Davis. The protocol was approved by the Committee on the Ethics of Animal Experiments of the University of California Davis (Animal Welfare Assurance Number A-3433-01). All procedures were performed under sodium pentobarbital anesthesia, and every effort was made to minimize suffering.

### Decision letter and Author response

Decision letter https://doi.org/10.7554/eLife.43322.015
Author response https://doi.org/10.7554/eLife.43322.016

## Additional files

### Supplementary files

• Supplementary file 1. Conversion efficiency of a selected subset of R-mAbs. Table details for a set of 181 projects (130 successful, 51 unsuccessful) the number of colony PCR or restriction digest positive clones that were input into the COS-ICC validation assay, and the number and percentage of these that were positive in that assay.
DOI: https://doi.org/10.7554/eLife.43322.011

• Supplementary file 2. mAbs converted to validated R-mAbs. Table lists the mAbs successfully converted to R-mAbs to date. For each mAb, the mAb clone number, the target protein, the mAb IgG subclass (in sentence case), the R-mAb IgG subclass (in upper case) and the original mAb and the cloned R-mAb RRID numbers in the Antibody Registry are detailed.
DOI: https://doi.org/10.7554/eLife.43322.012

• Transparent reporting form
DOI: https://doi.org/10.7554/eLife.43322.013

### Data availability

Plasmids and R-mAb sequences will be made available via Addgene (https://www.addgene.org/James_Trimmer/).

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
