## [Decision Letter]

Thank you for submitting your article "A toolbox of IgG subclass-switched recombinant monoclonal antibodies for enhanced multiplex immunolabeling of brain" for consideration by *eLife*. Your article has been reviewed by three peer reviewers, and the evaluation has been overseen by Richard Aldrich as the Senior Editor and Reviewing Editor. The following individuals involved in review of your submission have agreed to reveal their identity: Matthew N Rasband (Reviewer #1); Zoltan Nusser (Reviewer #2); Andrew Chalmers (Reviewer #3).

The reviewers have discussed the reviews with one another and the Reviewing Editor has drafted this decision to help you prepare a revised submission. The requested revisions are detailed in the individual reviews and should be straightforward to address.

*Reviewer #1:*

Much of the biomedical research community depends on the use of antibodies. The quality of the research they produce depends on the quality of the antibodies used. Over the last 25 years the Trimmer lab has been at the forefront of developing high quality and validated monoclonal antibodies for use in neuroscience research. For more than a decade this has been carried out at the NIH-funded neuromab monoclonal antibody resource at UC Davis. This resource has provided high quality, validated antibodies to thousands of scientists in the neuroscience community. This paper represents a remarkable additional step forward; not because of any conceptual advance, but because of a convergence of new technologies that the Trimmer lab has brought to bear on their extensive monoclonal antibody library. This technical advance makes it possible for the rapid and simple dissemination of high quality antibodies – of a variety of subclasses – to any laboratory. This makes it possible for any laboratory to simply acquire the plasmid for their antibody of interest from Addgene, then express the antibody in culture in their own lab. This paper also serves as a paradigm for how a lab can do this on their own, and how to apply the subclass switched R-mAbs. Broader acceptance and application of this strategy of making R-mAbs has the potential to dramatically improve the rigor and reproducibility, as well as reduce the overall costs associated with using antibodies in research.

I found the manuscript to be very straightforward, with enough detail for others to follow the overall strategy – as would be expected from a tools and resources paper. All experiments are very high quality. The examples of applications of the R-mAbs are also of very high quality. It still surprises me the number of scientists that do not understand how to multiplex experiments using different antibody subclasses. This paper emphasizes how the use of R-mAbs can dramatically and easily expand the repertoire of subclasses, effectively increasing the number of tools available to the investigator.

I think this paper is very appropriate for publication in *eLife* as a 'tools and resources' paper. I have no major suggestions – only one minor suggestion for improved presentation. The blue channels are very difficult to seeing Figures 3A, 3C, and 6A and 6B. Perhaps the authors could think about using gray scale for the blue channels?

*Reviewer #2:*

This manuscript describes a workflow of how to transfer monoclonal Abs (mAbs) to recombinant Abs (R-mAbs). In the past decades, the authors developed a large number of mAbs against various proteins expressed in the CNS, which became widely used tools in neuroscience. Most of their mAbs are extensively characterized in WB, ICC and IHC experiments. The authors provide a clear and convincing argument of why it is beneficial for the community to create R-mAbs from their mAbs. In the Results section, the authors describe many technical details regarding their cloning strategy and verifications. These are clearly essential, but the description in the results makes hard to read the manuscript. The reviewer suggests to remove some technical parts from the Results and put a reference to the Materials and methods. The part describing the generation of R-mAb from non-viable hybridoma is also too long and technical. It might have been technically challenging, but the important point is that it is possible with a similar strategy to that used for normal hybridomas.

The authors demonstrate that the specificity of the R-mAb is similar to the original mAb using WB, ICC and IHC. The presented data are very convincing. However, the reviewer would have liked to see some data regarding the affinity of the R-mAb. Determining the amount of IgG for the classical and the recombinant Abs should not be a problem and by creating a dilution line, the authors should demonstrate how the immunosignal depends on the Ab concentration.

The authors describe (subsection “Effective Cloning and IgG Subclass Switching of a Widely Used Monoclonal Antibody”, last paragraph) data shown in Figure 3D and E. There is no D and E panel in the 3rd figure! It would be very useful if the authors indicated not only the code number of the Abs in their images, but also the target antigen (e.g. in Figure 4A not only 'K58/35R', but 'K58/35R (pan-Nav)' and in all other figures).

Subsection “Multiplex brain immunofluorescent labeling with subclass switched R-mAbs”: Should not the 'N96/55' be 'N96/55R'?

*Reviewer #3:*

The work by Andrews and colleagues reports a set of methods to generate and validate recombinant monoclonal antibodies from existing libraries of hybridomas, including those which may no longer express the original clone. These methods were used to convert a widely used collection of hybridomas, the converted recombinant antibodies will be made available to the research community and the methods described could be carried out by most labs. This combination of new antibodies and new method has the potential to significantly promote the adoption of recombinant monoclonals which would help the research community due to their advantages, including greater transparency from the availability of the unambiguous sequence, easier archiving and sharing of reagents and improved reproducibility as they represent a molecularly defined reagent that should not be prone to genomic instability induced changes.

In addition recombinant monoclonals allow for conversion of the clones to new antibodies with distinct properties. This is nicely illustrated in this study as the methods developed allow for switching of the mouse IgG subclass, this makes multi-labelling experiments using subclass specific secondary antibodies possible that were not before, due to limitations in the availability of antibodies of different IgG subclasses. This if very nicely illustrated with a range of multi-labelling experiments.

Overall, I feel this work has the potential to support a wide range of research fields and warrants publication in *eLife*.

1) I think the panel labelling of Figure 3 might need correcting. The text and legend describes panels 3C-E, but the figure appears to only have panels 3A-C?

2) The authors state that plasmid sequences will be deposited in Addgene, once accepted for publication it would be good to add accession numbers to Table 3.

3) In the Introduction it states that most mAbs used for research are produced by hybridomas in culture. I do not believe this is quite accurate as many commercial suppliers have been actively switching to using recombinant antibodies, for example I believe almost all CST monoclonal antibodies are now produced recombinantly. I imagine methods from this study could help other suppliers switch their existing catalogues more efficiently.

---

## [Author Response]

Reviewer #1:[…] I think that the blue channels are very difficult to seeing Figures 3A, 3C, and 6A and 6B. Perhaps the authors could think about using gray scale for the blue channels?

We agree with the reviewer that the pure blue in these panels is difficult to see. We have revised the immunocytochemistry panels of these figures (3A, 6A, 6B) to make the tint of the blue much brighter by mixing equal parts blue and white. In our opinion this makes the details of the Hoechst labeling much easier to see without conflict with the red and green antibody labeling, but also retains the blue of the Hoechst dye itself. We changed the single-color blue panels on the immunoblots in panel 3C to pure greyscale, as retaining the blue was not as relevant here.

Reviewer #2:This manuscript describes a workflow of how to transfer monoclonal Abs (mAbs) to recombinant Abs (R-mAbs). In the past decades, the authors developed a large number of mAbs against various proteins expressed in the CNS, which became widely used tools in neuroscience. Most of their mAbs are extensively characterized in WB, ICC and IHC experiments. The authors provide a clear and convincing argument of why it is beneficial for the community to create R-mAbs from their mAbs. In the Results section, the authors describe many technical details regarding their cloning strategy and verifications. These are clearly essential, but the description in the results makes hard to read the manuscript. The reviewer suggests to remove some technical parts from the Results and put a reference to the Materials and methods.

We agree with the reviewer that while this is a toolbox article, some of the information in the Results section was redundant with and/or more appropriately presented in the Materials and methods section. As such we have removed/moved material from the Results section that we felt was in this category, such that the revised manuscript has a shorter Results section.

The part describing the generation of R-mAb from non-viable hybridoma is also too long and technical. It might have been technically challenging, but the important point is that it is possible with a similar strategy to that used for normal hybridomas.

We agree and reduced the size of this section.

The authors demonstrate that the specificity of the R-mAb is similar to the original mAb using WB, ICC and IHC. The presented data are very convincing. However, the reviewer would have liked to see some data regarding the affinity of the R-mAb. Determining the amount of IgG for the classical and the recombinant Abs should not be a problem and by creating a dilution line, the authors should demonstrate how the immunosignal depends on the Ab concentration.

We agree with the reviewer that this would be nice to show for one or more antibody pairs. In practice, this is more difficult to do accurately as in converting the native mAbs into R-mAbs, we also switched the IgG subclass. This results in a detection bias when probing these antibodies with the same generic secondary that sees all mouse subclasses, and even interferes with precisely determining antibody concentrations in conditioned media/tissue culture supernatants, and we do not yet have pure preps of the R-mAbs. As such while this goal seems easy to achieve in theory, it is more difficult to do well in practice. We are working towards reliable systems for being able to do this routinely, but it will be some time before we have this in place.

The authors describe (subsection “Effective Cloning and IgG Subclass Switching of a Widely Used Monoclonal Antibody”, last paragraph) data shown in Figure 3D and E. There is no D and E panel in the 3rd figure!

We apologize for this oversight and have corrected the text in the revised manuscript.

It would be very useful if the authors indicated not only the code number of the Abs in their images, but also the target antigen (e.g. in Figure 4A not only 'K58/35R', but 'K58/35R (pan-Nav)' and in all other figures).

This is a great suggestion and we have revised this figure to include target names.

Subsection “Multiplex brain immunofluorescent labeling with subclass switched R-mAbs”: Should not the 'N96/55' be 'N96/55R'?

We agree with the review that this sentence was ambiguous and have revised it accordingly.

Reviewer #3:1) I think the panel labelling of Figure 3 might need correcting. The text and legend describes panels 3C-E, but the figure appears to only have panels 3A-C?

We apologize for this oversight and have corrected the text in the revised manuscript.

2) The authors state that plasmid sequences will be deposited in Addgene, once accepted for publication it would be good to add accession numbers to Table 3.

We have in fact provided the first set of 92 R-mAb plasmids to Addgene and are preparing the remainder for shipment. However, the QA, production and QC process at Addgene is quite involved such that to date only 32 of the 92 we submitted have to date been made available. Included in the revised manuscript is a link (https://www.addgene.org/James_Trimmer/) to the section of the Addgene website that will catalog these, which includes the 32 that are currently listed and the others that will appear once they complete this process. We have not provided the ID numbers for these 32 as the information as of now remains fragmentary.

3) In the Introduction it states that most mAbs used for research are produced by hybridomas in culture. I do not believe this is quite accurate as many commercial suppliers have been actively switching to using recombinant antibodies, for example I believe almost all CST monoclonal antibodies are now produced recombinantly. I imagine methods from this study could help other suppliers switch their existing catalogues more efficiently.

We agree with the reviewer that recombinant mAbs (primarily rabbit) have an increased presence in the antibody world. However, many are still native from hybridomas (for example, all of these from DSHB). We have revised the Introduction from “most” to “many” to make this point.